# The nanophthalmos protein TMEM98 inhibits MYRF self-cleavage and is required for eye size specification

Sally H. Cross[1]*, Lisa Mckie[1], Toby W. Hurd[1], Sam Riley[1], Jimi Wills[1], Alun R. Barnard[2], Fiona Young[3], Robert E. MacLaren[2], Ian J. Jackson[1,4]

1 MRC Human Genetics Unit, MRC Institute of Genetics and Molecular Medicine, University of Edinburgh, Edinburgh, United Kingdom, 2 Nuffield Laboratory of Ophthalmology, Nuffield Department of Clinical Neurosciences, University of Oxford, The John Radcliffe Hospital, Oxford, United Kingdom, 3 Electron Microscopy, Pathology, Western General Hospital, Edinburgh, United Kingdom, 4 Roslin Institute, University of Edinburgh, Easter Bush, Midlothian, United Kingdom

* sally.cross@igmm.ed.ac.uk

## Abstract

The precise control of eye size is essential for normal vision. *TMEM98* is a highly conserved and widely expressed gene which appears to be involved in eye size regulation. Mutations in human *TMEM98* are found in patients with nanophthalmos (very small eyes) and variants near the gene are associated in population studies with myopia and increased eye size. As complete loss of function mutations in mouse *Tmem98* result in perinatal lethality, we produced mice deficient for *Tmem98* in the retinal pigment epithelium (RPE), where *Tmem98* is highly expressed. These mice have greatly enlarged eyes that are very fragile with very thin retinas, compressed choroid and thin sclera. To gain insight into the mechanism of action we used a proximity labelling approach to discover interacting proteins and identified MYRF as an interacting partner. Mutations of *MYRF* are also associated with nanophthalmos. The protein is an endoplasmic reticulum-tethered transcription factor which undergoes autoproteolytic cleavage to liberate the N-terminal part which then translocates to the nucleus where it acts as a transcription factor. We find that TMEM98 inhibits the self-cleavage of MYRF, in a novel regulatory mechanism. In RPE lacking TMEM98, MYRF is ectopically activated and abnormally localised to the nuclei. Our findings highlight the importance of the interplay between TMEM98 and MYRF in determining the size of the eye.

## Author summary

Having the correct eye size is important, too large and you will be short-sighted and too small and you will be far-sighted. Nanophthalmos, literally very small eye from the Greek, is a condition where the eye is very small but structurally normal. In addition to being far-sighted such eyes are prone to glaucoma which can lead to loss of vision. Here we studied a protein called TMEM98 which is found in the membranes of the cells which form a layer at the back of eye called the retinal pigment epithelium (RPE). Mutations in *TMEM98* have been found in nanophthalmos patients. Patients have one normal copy of

---

**Data Availability Statement:** The mass spectrometry proteomics data have been deposited to the ProteomeXchange Consortium via the PRIDE partner repository with the dataset identifier

---

PXD017091. All other relevant data are within the manuscript and its Supporting Information files.

**Funding:** SHC, LM, TWH, SR, JW and IJJ were funded by the MRC University Unit award to the MRC Human Genetics Unit, grant numbers MC_PC_U127561112 and MC_UU_00007/4. The funder website is https://mrc.ukri.org/. ARB and REM were funded by the NIHR Oxford Biomedical Research Centre. The funder website is https://oxfordbrc.nihr.ac.uk/. The funders had no role in study design, data collection and analysis, decision to publish, or preparation of the manuscript.

**Competing interests:** The authors have declared that no competing interests exist.

the gene and one carrying a mutation. We removed *Tmem98* from the RPE of mice in order to ascertain its function. We found, surprisingly, that rather than having small eyes this led to the development of very large eyes that were structurally fragile. We went on to identify protein partners of TMEM98 and found that it interacts with a protein called MYRF, mutations in which also cause nanophthalmos. This work demonstrates the importance of *TMEM98* in eye size specification.

## Introduction

Myopia and hyperopia impose a considerable disease burden in human populations and the incidence of refractive errors of the eye, myopia in particular, has increased greatly in the past few decades [1]. A critical factor in the development of refractive error is eye size, in particular axial length; too long and the eye is myopic, too short and it is hyperopic. Accurate control of eye growth is thus essential for normal vision. Whilst changes in human behaviour and environmental factors have played a part in the world-wide increase in myopia, genetic specification of eye size is also important. Many genes have been identified that are implicated in the occurrence of refractive error and thus of eye size [2–5]. Among these is *TMEM98*, mutations of which have also been found to be associated with dominant nanophthalmos [6, 7]. Variants close to the 5' end of *TMEM98* are associated with myopia in genome wide association studies [2, 4, 5], in which the minor alleles are associated with myopia and most likely a larger eye.

*TMEM98* encodes a highly conserved 226 amino acid transmembrane protein. It was initially reported to be part of an expression signature characteristic of adenocarcinoma, a subtype of adenosquamous carcinoma, a type of cervical cancer [8]. Subsequently it was suggested to be a novel chemoresistance-conferring gene in heptacellular carcinoma [9]. It has also been reported to be able to promote the differentiation of T helper 1 cells [10]. Studies using small interfering RNAs to knockdown expression of *TMEM98* have pointed to it having a role in the invasion and migration of lung cancer cells and in atherosclerosis [11, 12] and it may also be important for wound healing [13]. *TMEM98* was one of a set of genes used to construct a developmental hierarchy of breast cancer cells which could inform treatment strategies and disease prognosis [14].

However, two missense mutations in *TMEM98*, A193P and H196P, and a small deletion that removes the distal part of exon 4 and the beginning of the adjacent intron are associated with dominant nanophthalmos in humans [6, 7]. When we introduced the two human disease-associated missense mutations into the mouse gene we found no dominant phenotype [15]. However, when homozygous or as compound heterozygous these mutations caused a progressive retinal folding phenotype but we observed no significant change in eye size [15]. A different missense mutation in *Tmem98*, I135T, was found to cause a similar dominant retinal phenotype but was homozygous lethal [15]. The eyes of mice that are heterozygous for a knockout allele for *Tmem98* are normal showing that in the mouse haploinsufficiency for *Tmem98* does not cause an eye phenotype [15]. When pathogenic mutations in other genes causing nanophthalmos in humans have been studied in mice it has also been found that significant changes in eye size do not result, rather the predominant phenotype is retinal defects in line with what we found with *Tmem98* [16–27].

The homozygous lethality of a global *Tmem98* knockout precluded the possibility of studying the adult eye phenotype. To address the question of the importance of *Tmem98* expression in eye development we have knocked-out *Tmem98* specifically in the eye and find that an enlarged eye results. Both the phenotype of human patients and the mouse knockout

phenotype indicate the importance of *TMEM98* in eye size specification. We have gone on to examine the function of TMEM98 by identifying interacting proteins and find that it binds to, and prevents the self-cleavage, of the myelin regulatory factor, MYRF. The *MYRF* gene is itself associated with mutations leading to nanophthalmos [28–31]. Our studies confirm this pathway critical for eye size specification.

## Results

### Knockout of *Tmem98* causes eye enlargement, retinal and RPE abnormalities

*Tmem98* encodes a highly conserved 226 amino acid transmembrane protein. In the eye it is expressed during development in the RPE, and to a lesser extent in the ganglion cell layer (Fig 1A). In the adult eye it is highly expressed in the RPE, ciliary body and the iris (S1 Fig) [15].

As variants or mutations in human *TMEM98* are associated with larger or smaller eyes in humans we aimed to examine the role of the protein in eye development, and in particular the phenotype of a knockout of the gene in mice. First we examined the eyes of homozygous knockout mice at late stages of foetal development. At embryonic day (E)16.5-E17.5 the majority (5/6) knockout mice have eyes with an elongated shape when compared to wild-type and

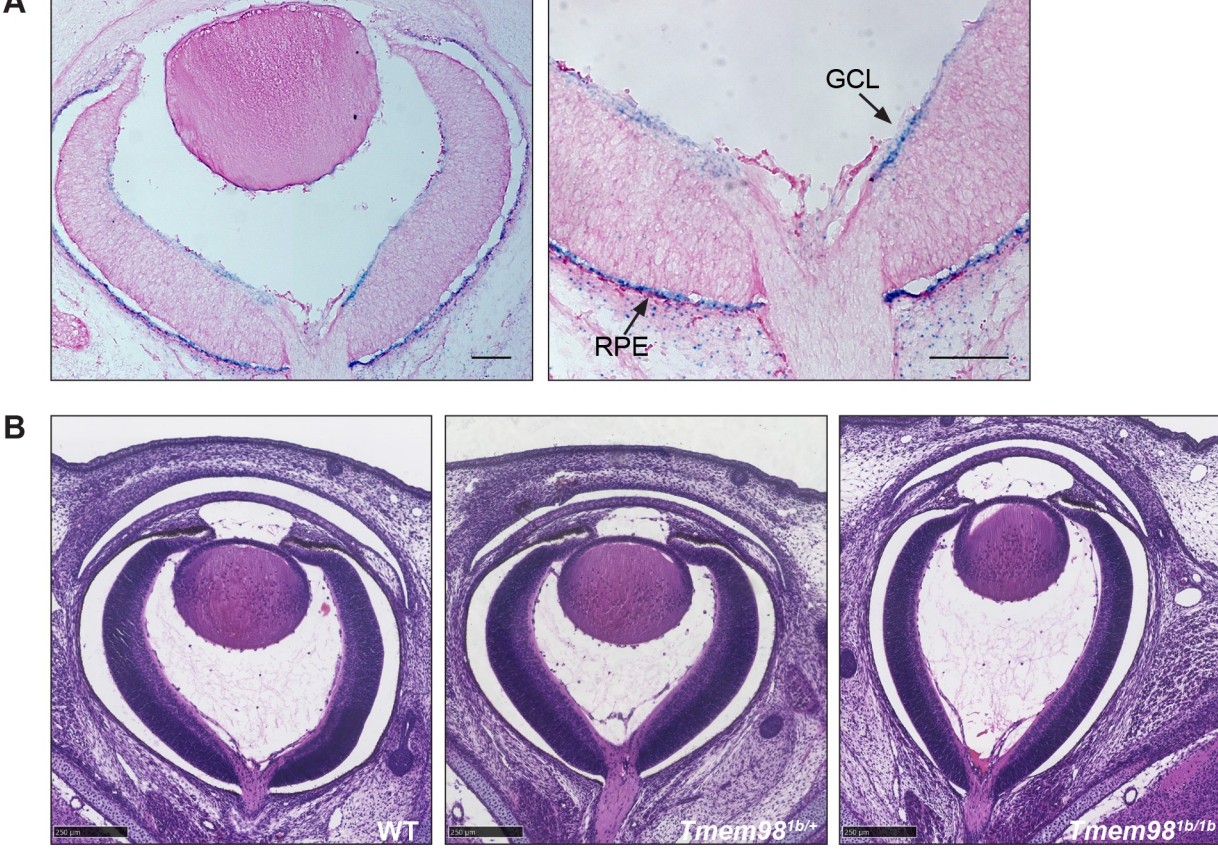

**Fig 1. Loss-of function of *Tmem98* results in an elongated eye shape.** (A) LacZ stained *Tmem98*^tm1b/+ E16.5 eye cryosections. The expression pattern of *Tmem98*, indicated by the blue staining for the reporter knockout allele *Tmem98*^tm1b, is found predominantly in the RPE and is also observed in the ganglion cell layer. (B) H&E stained eye sections of wild-type (left), *Tmem98*^tm1b/+ (centre) and *Tmem98*^tm1b/tm1b (right) E16.5 littermate embryos. The eye shape is elongated in the *Tmem98*^tm1b/tm1b embryo compared to the wild-type and heterozygous embryos. Abbreviations: GCL, ganglion cell layer and RPE, retinal pigment epithelium. Scale bars represent 100 μm (A), 250 μm (B).

heterozygous knockout eyes (Fig 1B), suggesting that loss of *Tmem98* impacts control of eye size. As complete knockout of *Tmem98* is perinatal lethal [15] to examine the phenotype of adult eyes in the absence of *Tmem98* we generated mice where *Tmem98* is knocked-out only in the eye. We converted the "knockout-first" allele *Tmem98^{tm1a}* to the conditional allele *Tmem98^{tm1c}* as described in Materials and Methods. In the presence of Cre the floxed critical exon 4 in the *Tmem98^{tm1c}* allele is removed, generating a deletion allele that has a frameshift, *Tmem98^{tm1d}*. The eyes of mice homozygous for the conditional allele, *Tmem98^{tm1c/tm1c}* (S2 Fig) and compound heterozygotes for the conditional with the deletion allele, *Tmem98^{tm1c/tm1d}* (Figs 2B, 2D and 3B) are normal. To knockout *Tmem98* in the RPE we used mice carrying the Tyr-Cre transgene, where Cre recombinase is expressed under the control of the tyrosinase promoter in the RPE and melanocytes derived from the neural crest [32, 33]. We verified where in the eye the Tyr-Cre transgene was expressed by examining mice that carried both the Tyr-Cre transgene and the R26MTMG transgene. The R26MTMG transgene expresses the tomato fluorescent protein ubiquitously under the control of the *Rosa26* promoter except in cells expressing Cre where the tomato is excised and green fluorescent protein is expressed instead [34]. This showed that the Tyr-Cre transgene is active in the RPE, ciliary body and iris (S3 Fig), all also sites of expression of *Tmem98* (Fig 1A and S1 Fig). By crossing *Tmem98^{tm1d/+}*; Tyr-Cre mice with *Tmem98^{tm1c/tm1c}* mice we generated *Tmem98^{tm1c/tm1d}*; Tyr-Cre mice and control littermates. As expected, because *Tmem98* expression was only absent in the eye, these conditionally knocked-out mice were viable and fertile. Unexpectedly, given the association between *Tmem98* and nanophthalmos, adult eyes in which *Tmem98* had been knocked-out by Tyr-Cre were greatly enlarged (Fig 2A–2C) and fundal imaging revealed that there was extensive retinal degeneration in the mutant eyes (Fig 2D). These abnormalities were not observed in mice with Tyr-Cre and at least one functional copy of *Tmem98* indicating that Cre expression alone did not elicit any defects (Fig 2B and 2D).

To further investigate the retinal phenotype we examined mutant and control eyes by scanning laser ophthalmoscopy (SLO) (Fig 2E). Infrared (IR) reflectance and multicolour mode imaging confirmed the observation of retinal degeneration in mutant eyes shown in colour fundus photographs; compared to the largely homogenous background of wild-type eyes, mutant eyes had a more disrupted and granular appearance. Blue autofluorescence (BAF) imaging showed conditional mutant retinas to have a reduction in the overall background level of autofluorescence but with a higher number of hyperfluorescent puncta compared to wild-type eyes, further confirming retinal degeneration and suggesting unhealthy and/or loss of RPE cells. Imaging with the near infrared autofluorescence (IRAF) mode showed that the overall level of signal was reduced in mutant eyes compared to wild-types, such that recording a reliable averaged image was not possible in the former.

## The RPE and retina are abnormal in the *Tmem98* conditional knockout

In RPE lacking *Tmem98* the cellular architecture is highly aberrant (Fig 3A). To study the morphology of the RPE we stained flat mount preparations of RPE for ZO1 (also known as TJP1) which is located at tight junctions between cells [35]. In the control, staining for ZO1 outlines the typical regular hexagonal shape of the RPE cells which contain either one or two nuclei (Fig 3A, left). In contrast, the mutant RPE lacking *Tmem98* has a highly irregular pattern of polygonal-sided cells which vary greatly in both their size and the number of sides and many are multinucleated (Fig 3A, right). The presence of these large irregular cells may indicate that cell death has occurred and that the remaining cells have enlarged and fused to compensate for cell loss and to maintain a continuous epithelial layer with no gaps which has been observed in a model of RPE cell ablation [36]. On histological examination the lens and anterior portion of

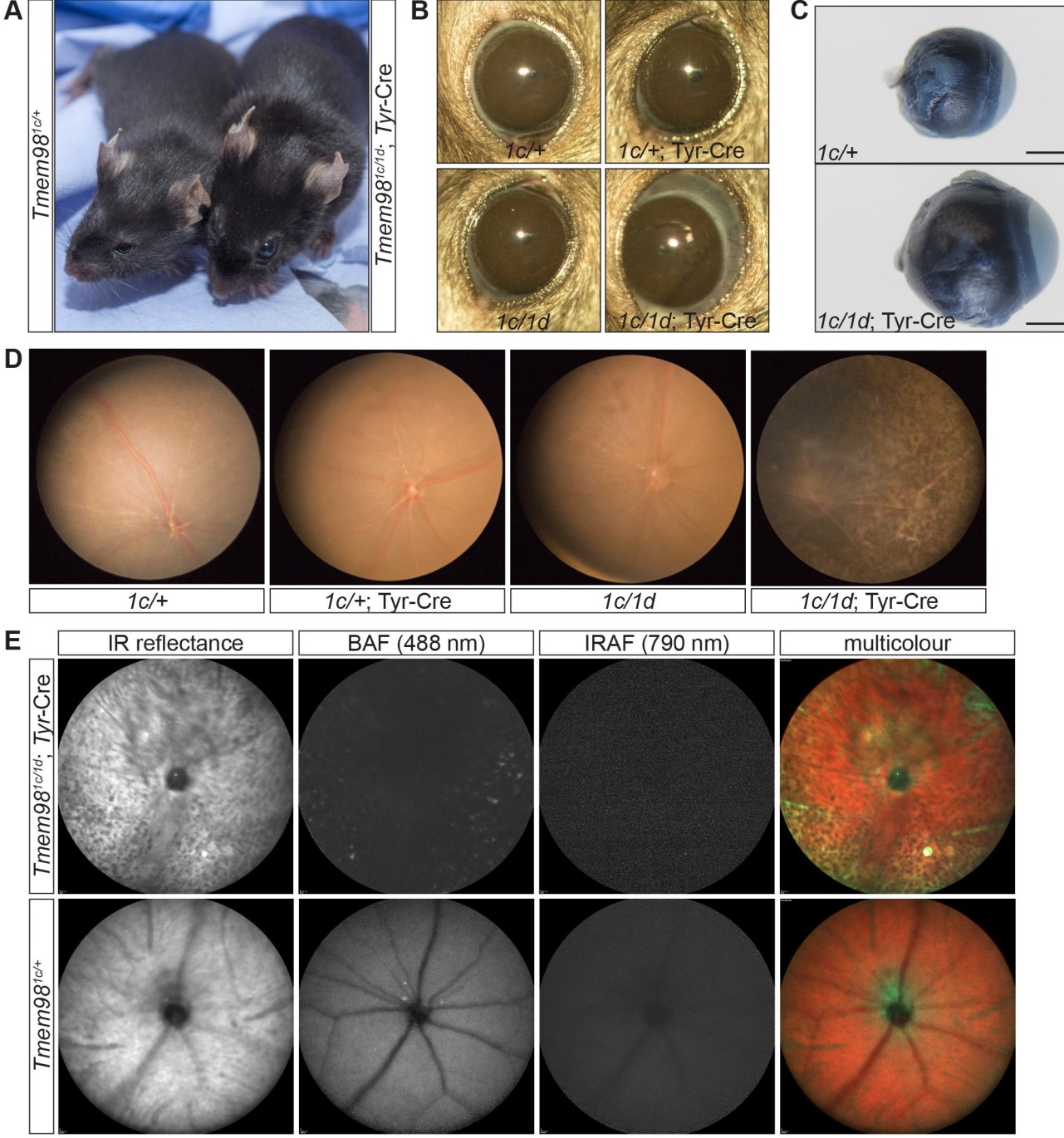

**Fig 2. Loss-of-function of *Tmem98* in the eye leads to an enlarged eye and retinal defects.** (A) *Tmem98^{tm1c/+}* (left) and *Tmem98^{tm1c/tm1d}*; Tyr-Cre (right) 9 week old female littermates are shown. When *Tmem98* is knocked-out by Tyr-Cre the eyes are noticeably enlarged. (B) Slit-lamp pictures of 9 week old littermates. The eyes of *Tmem98^{tm1c/+}* (female), *Tmem98^{tm1c/+}*; Tyr-Cre (male) and *Tmem98^{tm1c/tm1d}* (female) mice are normal indicating that haploinsufficiency for *Tmem98* and expression of Tyr-Cre does not affect eye size. In contrast the *Tmem98^{tm1c/tm1d}*; Tyr-Cre (female) eye, where *Tmem98* expression is lost, is enlarged and bulges out of the head. (C) Comparison of *Tmem98^{tm1c/+}* (top) and *Tmem98^{tm1c/+}*; Tyr-Cre (bottom) enucleated eyes. The eyes were collected from female littermates at three months of age. The posterior segment of the eye is enlarged in the *Tmem98^{tm1c/+}*; Tyr-Cre eye. Scale bar represents 1mm. (D) Fundus images of *Tmem98^{tm1c/+}* (female), *Tmem98^{tm1c/+}*; Tyr-Cre (male), *Tmem98^{tm1c/tm1d}* (female) and *Tmem98^{tm1c/tm1d}*; Tyr-Cre (female) mice. The pictures were taken at 7 weeks of age for the first three and at 10 weeks of age for the fourth. There is extensive retinal degeneration in the *Tmem98^{tm1c/tm1d}*; Tyr-Cre whilst the retinas of the other genotypes are normal. (E) Scanning laser ophthalmoscope images of control (*Tmem98^{tm1c/+}*) and mutant (*Tmem98^{tm1c/tm1d}*; Tyr-Cre) eyes from female littermates at 15 weeks of age. Abbreviations: IR, Infrared; BAF, blue autofluorescence and IRAF, near infrared.

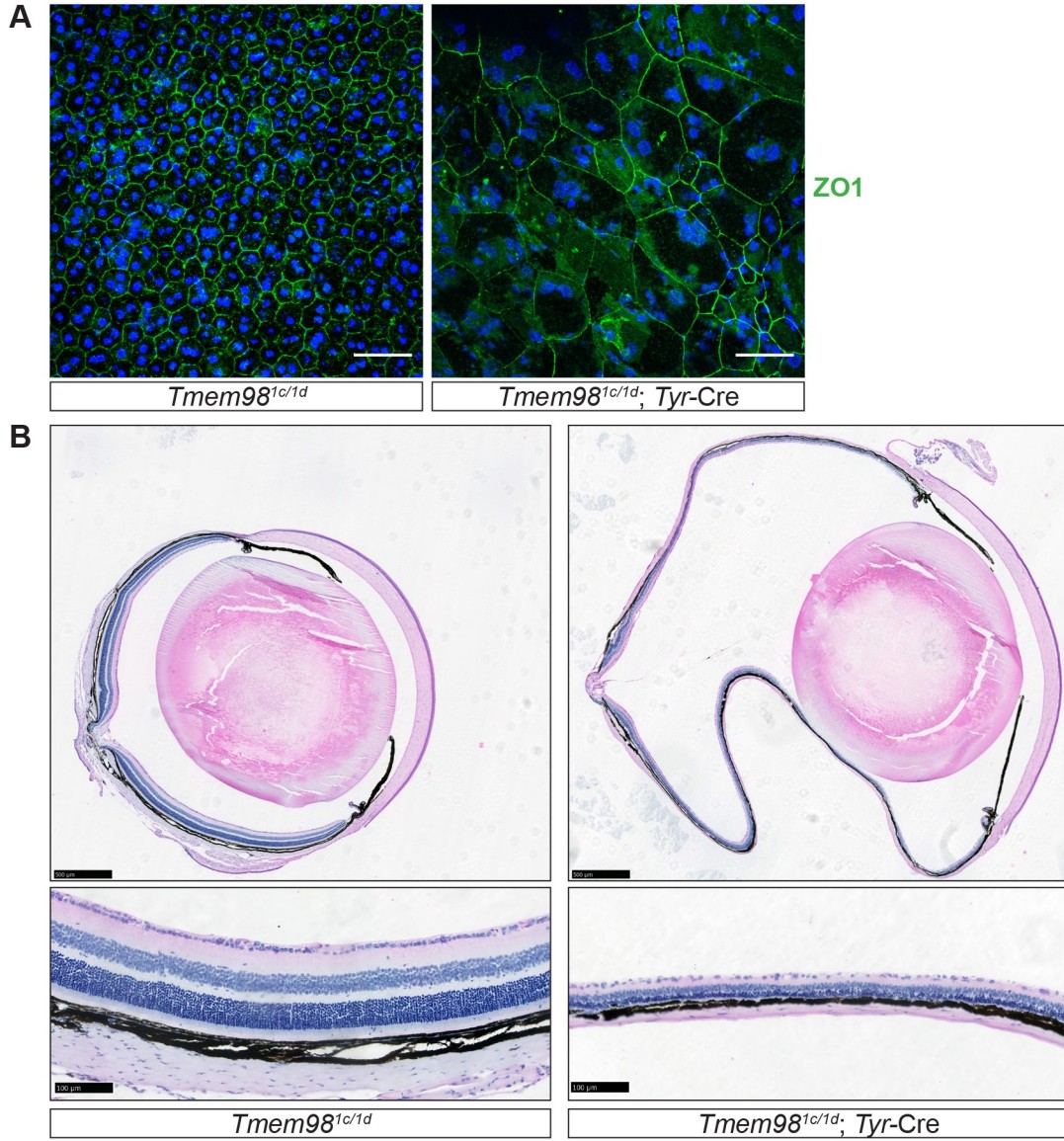

**Fig 3. Characterisation of the phenotype caused by loss-of-function of *Tmem98* in the eye.** (A) Staining of RPE flat mounts with an anti-ZO1 antibody (green) and DAPI (blue) reveals the regular hexagonal shape of the cells which contain one or two nuclei in the control *Tmem98^{tm1c/tm1d}* RPE. In contrast, the cells in the *Tmem98^{tm1c/tm1d}*; Tyr-Cre RPE vary greatly in size, the number of sides they have and many are multinuclear. The RPEs were collected from male littermates at 7 weeks of age. (B) H&E stained adult eye sections. The *Tmem98^{tm1c/tm1d}*; Tyr-Cre retina is hugely expanded and very thin compared to the control *Tmem98^{tm1c/tm1d}* retina. In addition, it appears to have lost structural integrity as indicated by the folding of the retina that occurred during processing which is not seen in the control. All the layers of the retina appear to be present (bottom) but are all extremely thin. In addition the choroid is compressed and the sclera is very thin. The eyes were collected from female littermates at 16 weeks of age. Scale bars represent 50 μm (A), 500 μm (B, top) and 100 μm (B, bottom).

eye appeared normal in eyes lacking *Tmem98* but the retina was massively elongated and the choroid layer was compressed (Fig 3B). The eye globe was extremely structurally fragile with little integrity such that the eye folded on histological preparation and did not retain the normal globular shape (Fig 3B). The sclera also appeared to be very thin but when we investigated it by transmission electron microscopy we did not observe any obvious abnormalities or changes in the collagen fibrils (S4 Fig). The retina was also extremely thin and whilst all the

layers of the retina appear to be present they were all reduced in thickness (Fig 3B). The increase in eye size appears to be restricted to the posterior, retinal segment. From histological sections, including the optic nerve, we measured the circumference of mutant and control eyes (S1 Dataset). The total circumference of the mutant eyes (N = 6) was greater than controls (N = 8) (mean control 8218+/-352 μm, mean mutant 12289+/-1378 μm P = 0.007, unpaired two-tailed *t*-test). The circumferential length of the anterior segments of both groups of eyes was not significantly different (mean control 3287+/-273 μm, mean mutant 3165+/-351 μm P = 0.79, unpaired two-tailed *t*-test) whereas the circumference of the posterior, retinal, segment of mutants was considerably greater (mean control 4931+/-215 μm, mean mutant 9124 +/-1058 μm P = 0.0008, unpaired two-tailed *t*-test). Furthermore, we measured corneal thickness both in histological sections and by optical coherence tomography (OCT) (S1 Dataset) and did not find a significant difference between controls and mutants, consistent with a lack of expansion of the anterior segment. We asked if the thinner retina was simply due to the expansion of the surface area of the globe. The increase in circumference of the retina corresponds to a ~2.9-fold increase in surface area (see Materials and Methods). By measuring the retina on the histological sections at six positions, three on each side of the optic nerve, we found the mean thickness of the mutant to be much smaller than controls (control 175.50 +/-12.05 μm, mutant 61.75 +/- 5.21 μm, 6 measurements on 4 eyes for both). The ratio of control to mutant thickness is 2.84, consistent with the reduction in thickness being due to the same number of retinal cells spread over a larger area, although we cannot rule out the possibility that some degeneration and cell death has occurred. The number of photoreceptors, as indicated by the number of nuclei spanning the outer layer have also correspondingly decreased (control 11.9 +/-0.55, mutant 4.8 +/-0.32, 6 measurements on 4 eyes). Again the control to mutant ratio of 2.48 indicates that there is very little, if any, loss of photoreceptors.

At E17.5 some, but not all conditionally knocked-out eyes (1/3) had a similar elongated shape as observed in the complete knockout (S5A Fig). By post-natal day (P)6 the increase in the size of the posterior segment and the thinning of the retina was apparent (S5B Fig), well before the eyes open at about P14, indicating that the disproportionate elongation and thinning of the retina mostly occurs in the immediate post-natal period.

## Retinal function is severely impaired in *Tmem98* conditional knockout eyes

To assess retinal function we carried out electroretinography on control (n = 3) and conditional mutant mice (n = 3) at 6 months of age (Fig 4). The electroretinogram (ERG) response was severely attenuated in the conditional mutant mice (Fig 4A). The amplitudes of both the a-wave, indicative of rod photoreceptor cell response to light stimulus, and the b-wave, indicative of the activation of Müller and bipolar cells were both greatly reduced (Fig 4A–4C). These results suggest that there is a much diminished retinal response to light by the remaining thin retina in the conditional mutant mice.

### *Tmem98* conditional knockout eyes show signs of retinal stress

We next investigated the pattern of expression of glial fibrillary acidic protein (GFAP). GFAP is an intermediate filament protein that undergoes upregulation in the Müller cells in response to retinal stress, injury or damage [37]. In *Tmem98*[tm1c/tm1d] and *Tmem98*[tm1c/+]; Tyr-Cre retinas we observed normal expression levels and localisation of GFAP whereas in the *Tmem98*[tm1c/tm1d]; Tyr-Cre retina GFAP expression is increased and can be seen extending down to the outer nuclear layer of the retina indicating a stress response (Fig 5A). We stained retinal sections for EZRIN, an actin-binding protein located in the apical microvilli of the RPE and TMEM98 (Fig 5B). In *Tmem98*[tm1c/tm1d] and *Tmem98*[tm1c/+]; Tyr-Cre retinas TMEM98 is

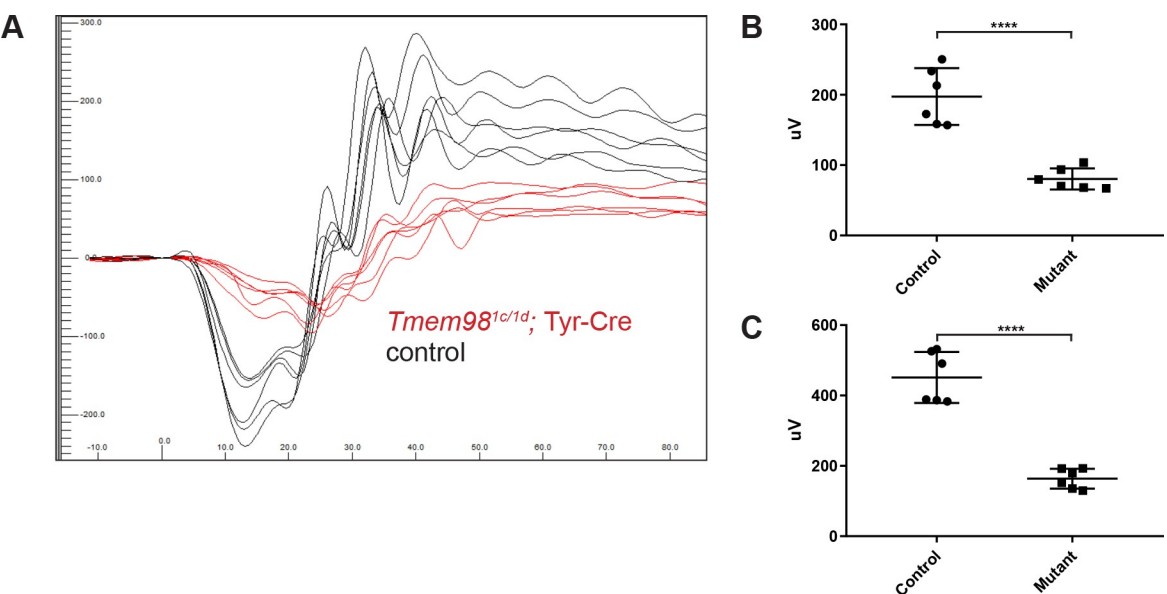

**Fig 4. Loss-of-function of *Tmem98* in the eye results in an attenuated ERG response.** Three *Tmem98*[tm1c/tm1d]; Tyr-Cre female mice and three control female mice (two *Tmem98*[tm1c/tm1d] and one *Tmem98*[tm1c/+]; Tyr-Cre) were tested at six months of age (S2 Dataset). (A) ERG traces of *Tmem98*[tm1c/tm1d]; Tyr-Cre mice (red lines) and control mice (black lines). Shown are the responses at 3 cd.s/m$^2$ (average of four flashes) for all eyes. (B) Comparison of a-wave amplitudes between the mutant *Tmem98*[tm1c/tm1d]; Tyr-Cre mice and control mice. There is a significant difference between mutant and control mice (unpaired *t*-test, P < 0.0001). (C) Comparison of b-wave amplitudes between the mutant *Tmem98*[tm1c/tm1d]; Tyr-Cre mice and control mice. There is a significant difference between mutant and control mice (unpaired *t*-test, P < 0.0001). **** indicates P < 0.0001.

present at both the apical and basolateral surfaces of the RPE. EZRIN staining can be seen at the apical surface of the RPE where it colocalises with TMEM98 as indicated by yellow staining at the apical surface of the RPE (Fig 5B). In the *Tmem98*[tm1c/ tm1d]; Tyr-Cre retina EZRIN is present but it is more diffuse and does not appear to be present in the apical microvilli. TMEM98 is not detected in the RPE confirming its deletion. We also stained retinal sections for the rod and cone opsins RHODOPSIN and ML-OPSIN (Fig 5C). The outer segments in the *Tmem98*[tm1c/ tm1d]; Tyr-Cre retina are considerably shortened in comparison the *Tmem98*[tm1c/ tm1d] and *Tmem98*[tm1c/+]; Tyr-Cre retinas but both RHODOPSIN found in the rods and ML-OPSIN found in the cones are still present.

## TMEM98 is a Type II transmembrane protein and interacts with MYRF

TMEM98 has been reported to be a single-pass type II transmembrane protein in which the C-terminal part is extracellular [10]. By Western blotting of ARPE-19 cells fractionated into the different cellular compartments we confirmed the membrane localisation of TMEM98 (Fig 6A). We confirmed the topology of the protein by assessing the accessibility to antibodies of N-terminal GFP and C-terminal V5 epitope tags fused to TMEM98 in transiently transfected NIH/3T3 cells (Fig 6B). In non-permeabilised cells only the V5 C-terminal tag was accessible confirming that the C-terminus is extracellular or luminal and the very short N-terminal end is cytoplasmic.

As a first step in determining the function of TMEM98 in the RPE and to understand why its absence results in eye expansion and thinning of the retina we sought to identify interacting proteins. To do this we carried out a proximity-dependent biotin identification (BioID) screen [38] in ARPE-19 cells. We chose to use this cell line because it is derived from retinal pigment epithelium and expresses *Tmem98* (Fig 6A). We made an expression construct where we fused

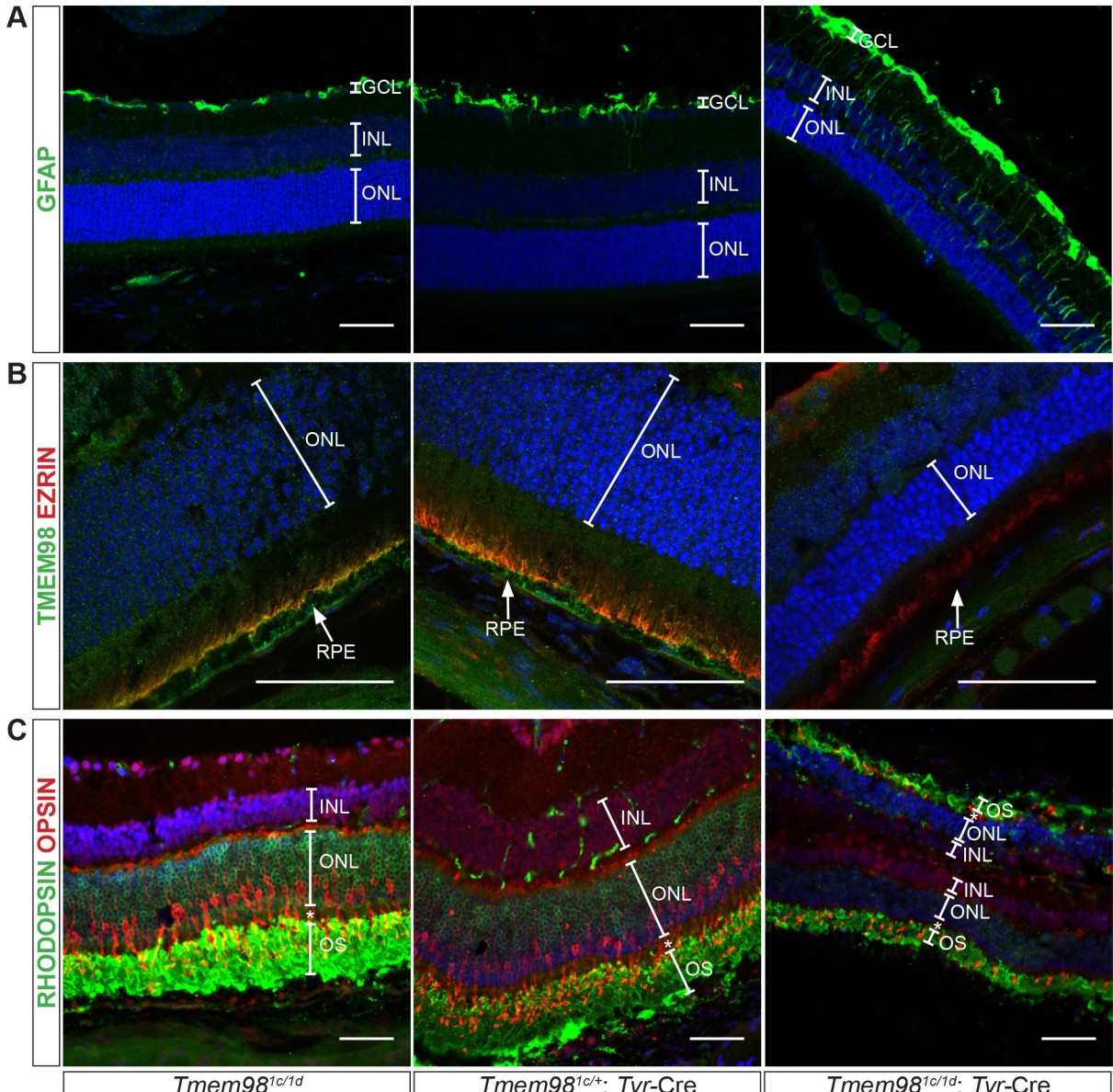

**Fig 5. Characterisation of the retinal phenotype caused by loss-of-function of *Tmem98*.** Immunostaining of adult retinal sections from control mice (*Tmem98^tm1c/tm1d* (left) and *Tmem98^tm1c/+*; Tyr-Cre (centre)) and mutant mice (*Tmem98^tm1c/tm1d*; Tyr-Cre (right)). (A) Staining with an anti-GFAP antibody (green) shows that GFAP localisation is normal and seen only in the ganglion cell layer in the control retinas but extends down towards the inner nuclear layer in the mutant retina indicating retinal stress. (B) Staining with anti-TMEM98 (green) and anti-EZRIN (red) antibodies. In the control retinas TMEM98 staining is seen in the apical and basal layers of the RPE and EZRIN is localised to the apical microvilli, co-localisation of the two proteins is indicated by yellow staining. In the mutant retina TMEM98 staining is absent and EZRIN appears to be mislocalised. (C) Staining with anti-RHODOPSIN (green) and anti-OPSIN (red) antibodies marking rods and cones respectively. Normal staining is observed in the control retinas but in the mutant although RHODOPSIN and OPSIN are present the outer segment layer is very thin. In the mutant the retina has folded back completely on itself. This is a processing artefact and reflects the fragility of the mutant retina. DAPI staining of DNA is in blue. Abbreviations: inner nuclear layer (INL), outer nuclear layer (ONL), outer segments (OS) and retinal pigment epithelium (RPE). The retinas shown in (A) and (B) were collected from P21 littermate male mice. The retinas shown in (C) were collected from 9 week old littermate male mice. Scale bars represent 50 μm.

a promiscuous biotin ligase (BirA*) domain to the C-terminal end of *Tmem98* and made a stably transfected ARPE-19 cell-line expressing this. We also made a stably transfected ARPE-19 control line expressing the BirA* domain alone. In the presence of biotin the BirA*

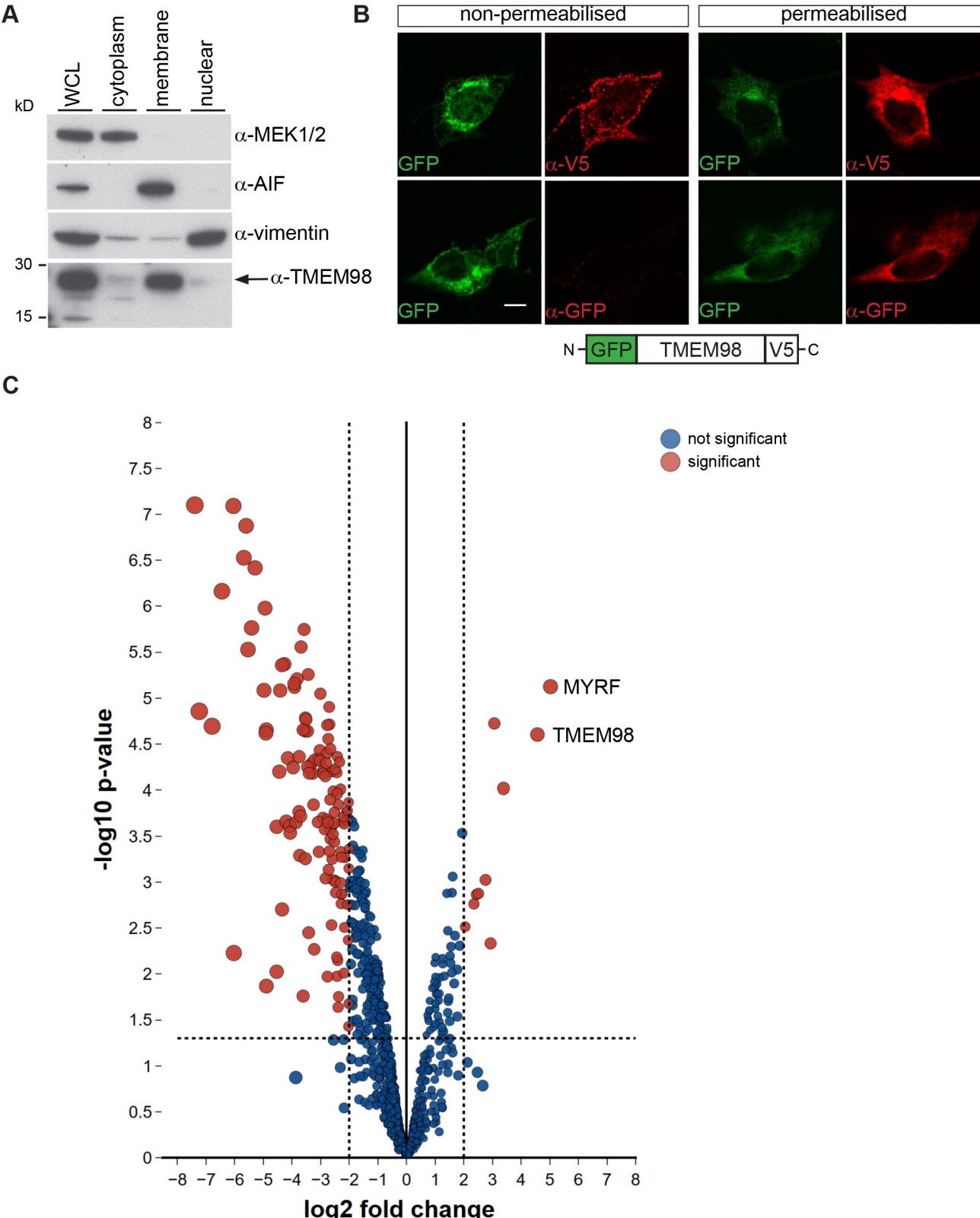

**Fig 6. TMEM98 is a type II transmembrane protein and interacts with MYRF.** (A) Western blot analysis of ARPE-19 subcellular fractions probed with the indicated antibodies. The control antibodies against MEK1/2, AIF and VIMENTIN are found in the expected fractions. TMEM98 is present in the membrane fraction. Uncropped Western blot images are shown in S8 Fig. WCL = whole cell lysate (B) Topology of TMEM98. NIH/3T3 cells were

transiently transfected with GFP-TMEM98-V5 (green) and immunostained with anti-GFP or anti-V5 antibodies (red). Non-permeabilised cells are shown on the left and permeabilised cells are shown on the right. On the top row transfected cells stained with an anti-V5 antibody are shown and on the bottom row transfected cells stained with an anti-GFP antibody are shown. Accessibility to the antibodies indicates that the GFP epitope is intracellular and that the V5 epitope is extracellular. A schematic of the expression GFP-TMEM98-V5 construct is shown below. (C) Volcano plot of the mass spectrometry data comparing proteins biotinylated by TMEM98-BirA* with those by BirA* alone. The x axis is log2 fold change of TMEM98-BirA* peptides versus BirA* peptides and the y axis is–log10 p-value of significance. The horizontal dashed line shows where p = 0.05 (−log10(0.05) = 1.3) and the vertical dashed lines show where fold change is four. Proteins with significant difference and change greater than four-fold are shown in red, other proteins are shown in blue. The two most highly enriched proteins, TMEM98 and MYRF, are labelled.

promiscuously biotinylates proteins within a 10 nm radius. We extracted protein from the cells, captured biotinylated proteins on streptavidin beads and subjected these to mass spectrometry analysis. We then compared the proteins isolated from the control and experimental cell lines. As expected TMEM98 was highly enriched in the experimental set indicating self-biotinylation (Fig 6C). A second protein, MYRF, had an even higher log2 fold change making it an excellent candidate for directly interacting with TMEM98 (Fig 6C). MYRF is an endoplasmic reticulum tethered transcription factor with a transmembrane domain near the C-terminal end of the protein. It assembles as a trimer and undergoes an autocatalytic cleavage reaction in an intramolecular chaperone domain that releases the N-terminal part of the protein which then translocates to the nucleus where it acts as a transcription factor important for oligodendrocyte specification and differentiation [39–42]. In all, 12 peptides derived from MYRF were found in the biotinylated fraction, nine from the N-terminal part and three from the C-terminal part. This suggested that either TMEM98 interacts with the full-length MYRF protein or with the two parts separately.

To test whether TMEM98 and MYRF directly interact we carried out co-immunoprecipitation experiments in HEK293T cells using TMEM98 tagged with V5 and MYRF, or a splice variant of MYRF lacking exon 19 (MYRFΔ19), tagged with MYC at the N-terminal end and FLAG at the C-terminal end. We tested the MYRFΔ19 splice form as well as the full-length as this is the predominant variant expressed in the RPE (B. Emery, personal communication and S6 Fig). When MYC-MYRF-FLAG or MYC-MYRFΔ19-FLAG was transfected alone the cleaved protein was detected by antibodies against MYC and FLAG but when transfected along with TMEM98-V5 the majority of the protein remained intact suggesting that TMEM98 inhibits the self-cleavage reaction (Fig 7A). After immunoprecipitation using anti-V5 antibodies both full-length and cleaved MYC-MYRF-FLAG or MYC-MYRFΔ19-FLAG could be detected by anti-FLAG antibodies, but only the full-length protein could be detected by anti-MYC antibodies suggesting that TMEM98 interacts with the C-terminal part of MYRF (Fig 7A). We also stained transfected cells with antibodies against the epitope tags. TMEM98-V5 localises to the membrane when transfected alone or with MYC-MYRF-FLAG (Fig 7B and 7D). When transfected alone MYC-TMEM98-FLAG undergoes cleavage and the N-terminal part translocates to the nucleus and the C-terminal part colocalises with the membrane (Fig 7C). When transfected along with TMEM98-V5 the N-terminal portion of MYC-MYRF-FLAG is no longer detected in the nucleus but is found in the membrane along with the C-terminal portion and they colocalise with TMEM98-V5 (Fig 7D). We also tested two missense mutants of *Tmem98*, I135T and A193P, to ascertain if the ability of TMEM98 to inhibit MYRF self-cleavage was compromised by these mutations in co-transfection experiments (S7 Fig). The I135T missense mutation of *Tmem98* causes folds in the retinal layers which is accompanied by a retinal white spotting phenotype when heterozygous and is lethal when homozygous [15] and the A193P missense mutation is associated with dominant nanophthalmos in humans [6] and causes a recessive retinal phenotype in the mouse that is similar to the dominant phenotype caused by I135T [15]. Neither of these missense mutations affected the ability of

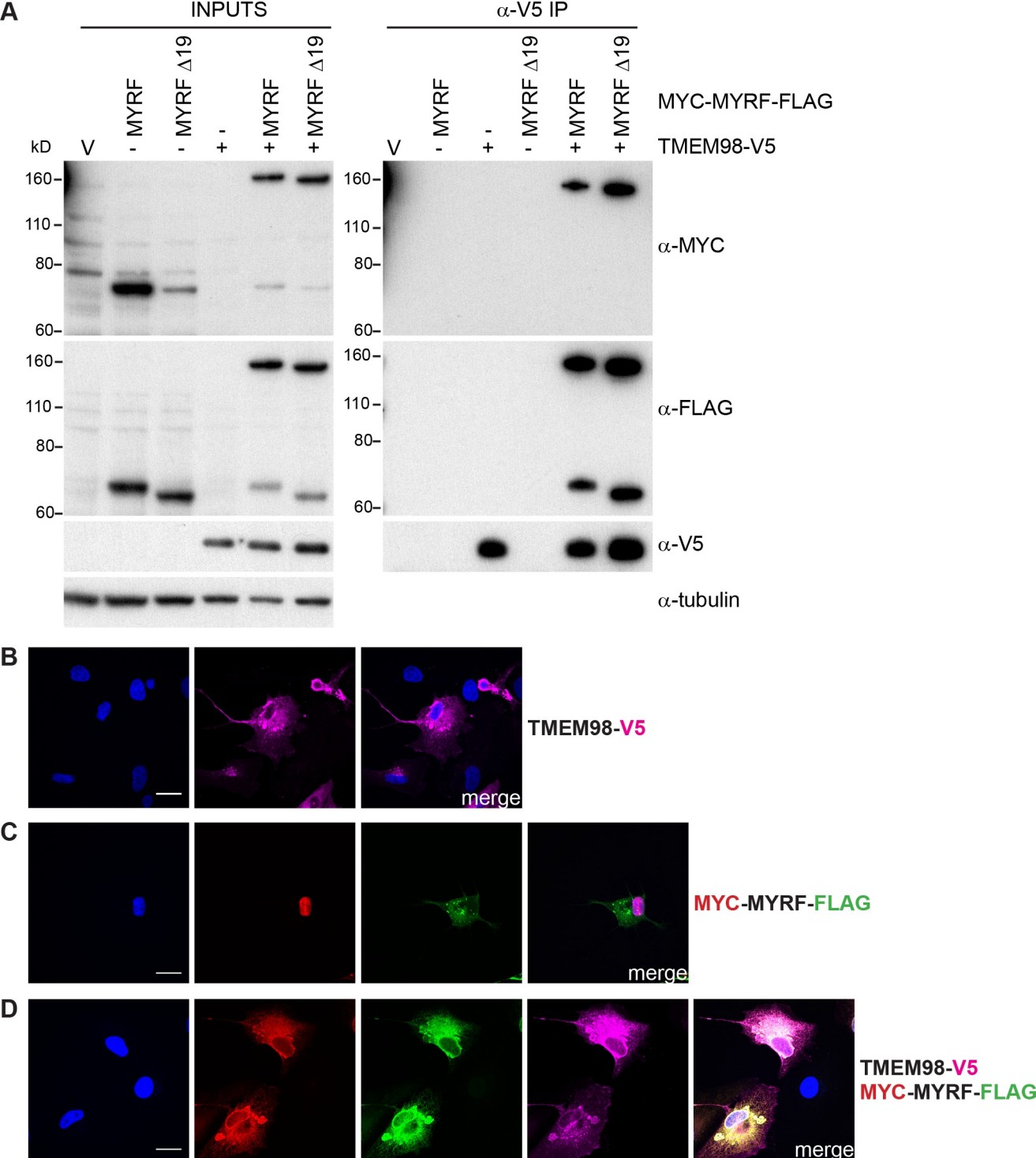

**Fig 7. TMEM98 prevents MYRF self-cleavage and binds to the C-terminal part of MYRF.** (A) Co-immunoprecipitation experiment where HEK293T cells were transiently transfected with the indicated epitope-tagged expression constructs and immunoprecipitated with anti-V5.The two MYRF constructs were either full-length (MYRF) or lacked exon 19 (MYRF Δ19). Western blot analysis of the inputs (left) and immunoprecipitated fractions (right) probed with anti-MYC (Cell Signaling Technology, 2276), anti-FLAG (Cell Signaling Technology, 2368) and anti-V5 antibodies are shown. Anti-tubulin antibody was used to probe the input Western as a loading control. The Western of the input samples shows that MYRF cleaves when transfected alone but remains largely intact when co-transfected with TMEM98-V5. The Western of the immunoprecipitated fractions shows that intact MYC-MYRF-FLAG and the C-terminal part tagged with FLAG are co-immunoprecipitated with TMEM98-V5 indicating that TMEM98 interacts with the C-terminal part of MYRF. Uncropped Western blot images are shown in S9 Fig. (B-D) ARPE-19 cells were transiently transfected with TMEM98-V5 and/or MYC-MYRF-FLAG and

immunostained with anti-V5 (magenta), anti-MYC (Cell Signaling Technology, 2278) (red) and anti-FLAG (Biolegend, 637302) (green) antibodies as indicated. DAPI staining is in blue. (C) When transfected alone MYC-MYRF-FLAG cleaves and the N-terminal part tagged with MYC translocates to the nucleus whilst the C-terminal part tagged with FLAG is membrane-bound. (D) When MYC-MYRF-FLAG is co-transfected with TMEM98-V5 it remains intact and colocalises with TMEM98-V5 in the membrane. Scale bars represent 20 μm.

TMEM98 to inhibit MYRF self-cleavage to any noticeable degree suggesting that the mutant phenotypes caused by these missense mutations are probably not due to a change in the interaction between TMEM98 and MYRF (S7 Fig).

## MYRF is mislocalised in *Tmem98* Conditional Knockout Eyes

Both *Tmem98* and *Myrf* are reported to be highly expressed in the human and mouse RPE (http://www.biogps.org). We have confirmed the high expression of Tmem98 in the RPE (shown in Fig 1A, Fig 5 and S1 Fig) [15]. We anticipated that in mutant RPE, MYRF would be abnormally activated, and liberate the nuclear-localised portion of the protein. To examine the localisation of MYRF in normal RPE and RPE devoid of TMEM98 we stained RPE flat-mount preparations from *Tmem98$^{1c/1d}$* and *Tmem98$^{1c/1d}$*; Tyr-Cre mice with antibodies directed against either the N-terminal or the C-terminal part of MYRF (Fig 8). In *Tmem98$^{1c/1d}$* RPE we observed that the N-terminal portion of MYRF could be detected both in the membrane and the nucleus (Fig 8, top) and the C-terminal portion was found principally in the membrane. As we had previously shown by ZO1 staining (Fig 3A) the architecture of *Tmem98$^{1c/1d}$*; Tyr-Cre RPE lacking *Tmem98* is highly abnormal and the pattern of localisation of MYRF was altered with the increased immunostaining for the N-terminal portion of MYRF in the nucleus (Fig 8, bottom).

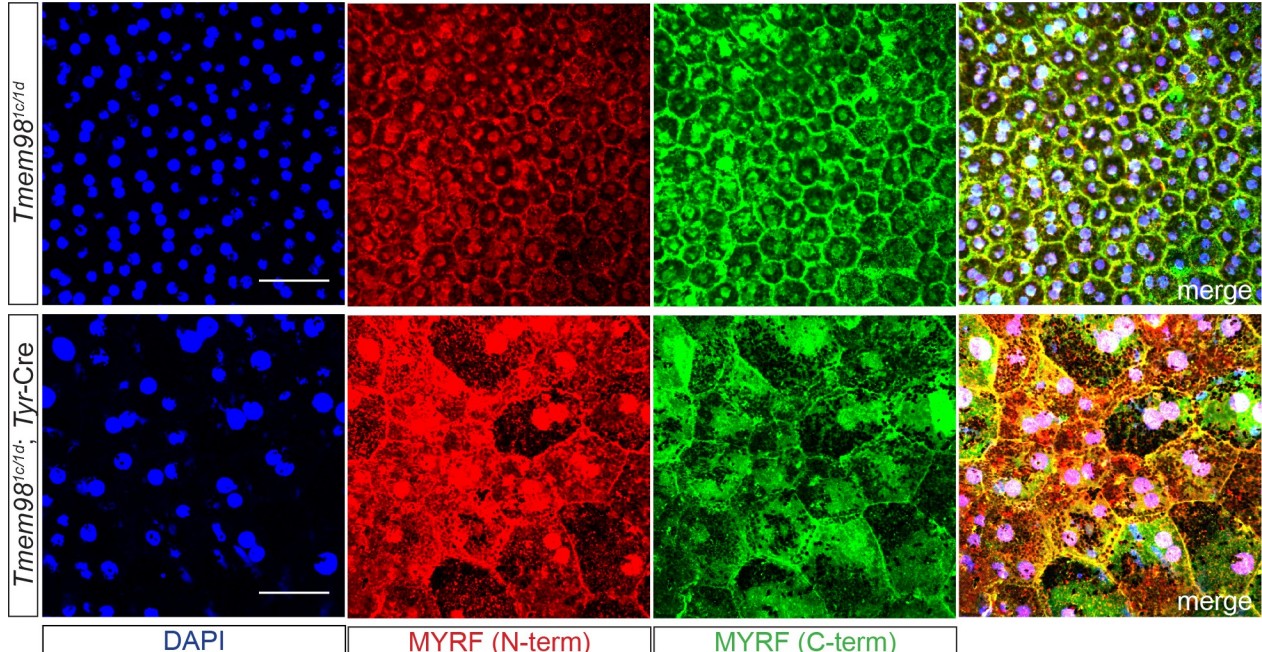

**Fig 8. MYRF is mislocalised in the RPE when *Tmem98* is knocked-out in the eye.** Immunostaining of control Tmem98$^{tm1c/tm1d}$ (top) and mutant Tmem98$^{tm1c/tm1d}$; Tyr-Cre RPE flat mounts with DAPI (blue), anti-MYRF N-terminal (red) and anti-MYRF C-terminal (green) antibodies. MYRF staining is aberrant and when compared to the control more of N-terminal part of MYRF appear to be present in the nucleus in the mutant when compared to the control. The RPEs were collected from P26 male littermate mice. Scale bars represent 50 μm.

## Discussion

In humans, mutations of *TMEM98* appear to cause dominant nanophthalmos [6, 7], whilst in population studies variants 5' of this gene are associated with myopia [2, 4, 5]. One of these variants, rs10512441, found 15 kb upstream of *TMEM98*, is located in a region of active chromatin likely to have regulatory function [43]. We have shown here that loss of TMEM98 protein in the mouse, and specifically in the RPE, results in a greatly enlarged eye and a proportionally thinner retina, compressed choroid layer and thin sclera. This excessive growth of the eye initiates before the mouse eyes are open, suggesting that the pathological changes we observe are not a result of the disruption of emmetropisation, the post-natal mechanism of eye growth and corneal changes that in concert ensure that light is focused perfectly on the retina. The increased eye size is restricted to the posterior, retinal portion, leaving the anterior segment essentially unchanged. The posterior portion is enclosed by the RPE, and numerous studies point to the interaction between RPE and the surrounding sclera to be necessary for control of eye growth [44–47]. However the mechanism by which the eye becomes enlarged is not simply loss of RPE function. Mutations which affect RPE function do not usually lead to alterations in eye size, rather the usual outcome is degeneration of the photoreceptors, for example loss-of-function of *RPE65* can lead to retinitis pigmentosa or leber congenital amaurosis [48, 49] and mutations in *C1QTNF5* can result in late-onset retinal macular degeneration [50]. Surprisingly, in our model we do not see substantial loss of photoreceptors. However, the reduced ERG response of the eyes where *Tmem98* has been knocked out indicates that they may have diminished function. The fragility of the mutant eyes during processing for histology suggests that the sclera is greatly weakened. However, on examination of the sclera by transmission electron microscopy we do not observe any obvious abnormalities (S4 Fig). Given the fragility of the eye a more detailed analysis of the sclera is merited in future studies. Nanophthalmos is associated with a thickened sclera, and it is possible that the cause of *TMEM98* associated nanophthalmos involves misregulation of the same pathway which leads to weaker sclera in this loss-of-function model.

Loss-of-function of *Tmem98* in the RPE results in eye enlargement whereas missense mutations that cause dominant nanophthalmos in humans result in retinal defects when homozygous in the mouse but no significant change in eye size [15]. We proposed that the mode of action of the missense mutations was a dose-dependent gain-of-function because compound heterozygous mice with one missense allele and one deletion allele are viable with no eye phenotype. The divergent consequences of mutations or variants on eye size is seen at other loci. *PRSS56* is another human gene in which mutations can cause recessive nanophthalmos [26, 51, 52] and variants of this gene are also associated with myopia [2, 3].

The interaction between TMEM98 and MYRF has been shown by others [28, 53]. We confirmed the interaction using a proximity modification technique, which does not necessarily rely on direct interaction, but we went on to demonstrate a direct interaction does indeed occur (Fig 7). The interaction inhibits the self-cleavage of MYRF and causes retention in the membrane, and is a novel mechanism of regulation. We can show that in the absence of TMEM98, MYRF is ectopically activated in the RPE and this may lead to aberrant gene regulation underlying the large eye phenotype. Interestingly, heterozygous mutations of *MYRF* in humans lead to nanophthalmos [28–31], suggesting loss of this protein has the opposite effect to the activation we suggest occurs in the *Tmem98* deletion eyes. However, no associations have been described in genome wide association studies between *MYRF* and myopia or any other eye disorder. RPE-specific deletion of MYRF in the mouse RPE does not lead to change in eye size but rather results in depigmentation of the RPE [28]. This study also found that when *Myrf* expression is reduced in the eye a reduction in *Tmem98* transcription results

accompanied by severely reduced detection of TMEM98 protein in the RPE. In spite of this depletion of TMEM98 the architecture of the RPE is unaltered in stark contrast to the highly aberrant RPE in the *Tmem98$^{tm1c/tm1d}$*; Tyr-Cre mice where only *Tmem98* is deleted (Fig 2D and Fig 3A). This suggests that in our experiment the eye size phenotype associated with loss of TMEM98 requires functional MYRF, further implicating dysregulation of MYRF in the mechanism and that the sequestration of MYRF at the endoplasmic reticulum occurs under physiological conditions and is not an artefact caused by overexpression. It would be interesting to determine the eye phenotype when *Myrf* is deleted as well as *Tmem98* in the RPE. A prediction might be that the enlarged eye phenotype would be rescued.

In summary, mouse mutagenesis provides a useful means of validating candidate genes identified through GWAS studies, and addressing the mechanism of action and pathways involved.

## Materials and methods

### Ethics statement

All experiments with mice complied with United Kingdom Home Office regulations. They were performed under United Kingdom Home Office project licence approval and were done in accordance with the ARVO Statement for the Use of Animals in Ophthalmic and Vision Research.

### Mice

Clinical examinations were carried out as previously described [54]. Fundus imaging was performed as described [55]. Mice carrying a targeted knockout-first conditional-ready allele of *Tmem98*, *Tmem98$^{tm1a(EUCOMM)Wtsi}$* (hereafter *Tmem98$^{tm1a}$*), were obtained from the Sanger Institute. By crossing *Tmem98$^{tm1a/+}$* mice with mice carrying a germ-line Cre the "knockout-first" *tm1a* allele is converted to the reporter knockout allele *Tmem98$^{tm1b(EUCOMM)Wtsi}$* (hereafter *Tmem98$^{tm1b}$* or *Tmem98$^{1b}$*). By crossing *Tmem98$^{tm1a/+}$* mice with mice carrying FLPe the *Tmem98$^{tm1a}$* 'knockout-first' allele was converted to the floxed conditional allele *Tmem98$^{tm1c(EUCOMM)Wtsi}$* (hereafter *Tmem98$^{tm1c}$* or *Tmem98$^{1c}$*). By crossing *Tmem98$^{tm1c/+}$* mice with mice carrying Cre the floxed critical exon 4 is deleted in cells expressing Cre generating a deletion allele that has a frame shift *Tmem98$^{tm1d(EUCOMM)Wtsi}$* (hereafter *Tmem98$^{tm1d}$* or *Tmem98$^{1d}$*) that would be subject to nonsense mediated decay [56]. FLPe expressing mice were made by Andrew Smith, University of Edinburgh [57]. Germ-line Cre expressing mice were made by Dirk Kleinjan, MRC Human Genetics Unit [58]. Tyr-Cre expressing mice were originally obtained from Lionel Larue, Institut Pasteur [33] and R26MTMG reporter mice were originally obtained from Liqun Luo, Stanford University [34]. Mice were initially genotyped by PCR using the primers listed in S1 Table and subsequently by Transnetyx (Cordova, TN, USA) using custom designed assays (http://www.transnetyx.com). All mouse lines were maintained on the C57BL/6J mouse strain background.

### Electroretinography

Before testing mice were dark-adapted overnight (>16 hours) and electroretinography was carried out in a darkened room under red light illumination using an HMsERG system (Ocuscience, Henderson, MV, US). Mice were anaesthetised with isofluorane and their pupils dilated with 1% tropicamide. Three grounding electrodes were placed subcutaneously, in the base of the body above the tail and each cheek, and silver-coated electrodes were positioned concentrically on the corneas with hypromellose eye drops (2.5% methylcellulose coupling

agent) held in place with contact lenses. Mice were placed on a heated platform to maintain their body temperature at 37˚C and monitored with a rectal thermometer. A modified Quick-RetCheck (Ocuscience) protocol was used to obtain full-field scotopic ERGs. Briefly, four flashes at 10 mcd.s/m$^2$ at two second intervals were followed by four flashes at 3 mcd.s/m$^2$ at ten second intervals and then by four flashes at 10 mcd.s/m$^2$ at ten second intervals.

## Confocal scanning laser ophthalmoscope (cSLO) and optical coherence tomography (OCT) imaging

Imaging was performed using a Spectralis OCT plus MultiColor, confocal scanning laser ophthalmoscope (Heidelberg Engineering, Heidelberg, Germany). Mice were anesthetised via intraperitoneal injection of ketamine (80 mg/kg body weight, Vetalar; Zoetis UK Limited, Leatherhead, Surrey, UK) and xylazine (10 mg/kg body weight, Rompun; Bayer PLC, Reading, Berkshire, UK) diluted in sterile 0.9% saline solution. Pupils were dilated with tropicamide 1% and phenylephrine hydrochloride 2.5% eye drops (both Bausch & Lomb, Kingston upon Thames, UK). In wild-type mice, a custom-made polymethylmethacrylate (PMMA) contact lens was placed on the cornea with 0.3% methylcellulose gel as viscous coupling fluid. SLO images were recorded using IR (infrared) reflectance, blue-light autofluorescence (BAF), IR autofluorescence (IRAF) and multicolour reflectance imaging modes. IR reflectance was used as a reference imaging to perform spectral-domain (SD)-OCT. The SD-OCT scans were acquired using volume protocols provided by the Spectralis software, orientated such that their superior and inferior sectors were positioned over the optic nerve head. Total retinal thickness was automatically calculated using the Heidelberg software. The longer/larger eye size in mutant mice meant that images could not be recorded using the same custom-made PMMA contact lens as above. Instead a flat, glass, 6mm-diameter round coverslip was used on a drop of 2% w/w polyacrylic acid gel (Viscotears, Bausch & Lomb, Kingston upon Thames, UK) placed on the cornea. Images were recorded using "automatic real time" (ART) mode, which tracks ocular movement (e.g., due to respiration) and averages consecutive images resulting in an improved signal-to-noise ratio. At least 25 images/OCT b-scans were averaged, except in the case of IRAF mode imaging in mutant mice, where the signal was too low to mediate tracking, and a single image was used.

## Antibodies

Details of primary antibodies used are given in Table 1.

## Histology and immunohistochemistry

Mice were culled by cervical dislocation, eyes enucleated and fixed in Davidson's fixative (28.5% ethanol, 2.2% neutral buffered formalin, 11% glacial acetic acid) for 1 hour at room temperature (cryosectioning) or more than 16 hours at 4˚C (wax embedding). Embryos, following dissection and removal of a tail sample to be used for genotyping, were fixed in Davidson's fixative at 4˚C. Before wax embedding fixed tissue was dehydrated through an ethanol series. Haematoxylin and eosin (H&E) staining was performed on 5–10 μm paraffin embedded tissue sections using standard procedures and images captured using a Nanozoomer XR scanner (Hamamatsu, Hamamatsu City, Japan) and viewed using NDP.view2 software (Hamamatsu). For cryosectioning, fixed tissue was transferred to 5% sucrose in phosphate buffered saline (PBS) and once sunk transferred to 20% sucrose in PBS overnight. Samples were then embedded in OCT compound on dry ice and cryosectioned at 14 μM. For immunostaining on cryosections, slides were washed with water and post-fixed in acetone at -20˚C for 10 minutes. They were then rinsed with water, blocked in 10% heat-inactivated donkey serum (DS) in

**Table 1. Primary antibodies.**

| Antibody | Source | Product No | Concentration used |
|---|---|---|---|
| anti-AIF | Cell Signaling Technology | 5318 | 1:5000 (WB) |
| anti-Ezrin | abcam | ab4069 | 1:100 (IF) |
| anti-FLAG | Cell Signaling Technology | 2368 | 1:2000 (WB) |
| anti-FLAG | BioLegend | 637302 | 1:1000 (IF) |
| anti-GFAP | abcam | ab7260 | 1:500 (IF) |
| anti-GFP | Molecular Probes | A-11122 | 1:1000 (IF) |
| anti-MEK1/2 | Cell Signaling Technology | 8727 | 1:5000 (WB) |
| anti-MYC | Cell Signaling Technology | 2276 | 1:2000 (WB) |
| anti-MYC | Cell Signaling Technology | 2278 | 1:200 (IF) |
| anti-MYRF (C-terminal) | gift from B. Emery [39] | - | 1:500 (IF) |
| anti-MYRF (N-terminal) | gift from M. Wegner [63] | - | 1:500 (IF) |
| anti-Opsin, red/green | Millipore | AB5405 | 1:500 (IF) |
| anti-Rhodopsin | Millipore | MAB5356 | 1:500 (IF) |
| anti-TMEM98 | proteintech | 14731-1-AP | 1:5000 (WB), 1:200 (IF) |
| anti-$\alpha$-tubulin | Sigma-Aldrich | T5168 | 1:10,000 (WB) |
| anti-V5 | Invitrogen | 46–0705 | 1:5000 (WB) (IF, Fig 6B), 1:400 (IF, Fig 7) |
| anti-vimentin | Cell Signaling Technology | 5741 | 1:5000 (WB) |
| anti-ZO1 | Thermo Fisher Scientific | 33–9100 | 1:100 (IF) |

Key: WB = Western blotting, IF = immunofluorescence

Tris-HCl buffered saline (TBS) with 0.1% Tween-20 in (TBST) for one hour and then incubated with primary antibodies diluted in TBST with 5% DS for two hours at room temperature or overnight at 4˚C. Subsequently, after washing with TBST, the slides were incubated with Alexa Fluor secondary antibodies (Invitrogen, Carlsbad, CA, USA) diluted 1:400 in TBST with 5% DS at room temperature for one hour. Following washing with TBST coverslips were mounted on slides in Prolong Gold (Thermo Fisher Scientific, Waltham, MA, USA) and confocal images acquired on a Nikon Confocal A1R microscope (Nikon Instruments, Tokyo, Japan). The microscope comprises of a Nikon Eclipse TiE inverted microscope with Perfect Focus System and is equipped with 405nm diode, 457/488/514nm Multiline Argon, 561nm DPSS and 638nm diode lasers. Data were acquired using NIS Elements AR software (Nikon Instruments). Images were processed using either NIS-Elements or ImageJ software [59].

## Eye size measurements

The circumference of control and mutant eyes was measured from H&E stained histological sections through the optic nerve using NDP.view2 software (Hamamatsu). To compare the retinal surface areas we treated the eye as a sphere, and calculated the radius, R, and the total surface area from the circumference. We then calculated the surface area of the anterior (non-retinal) portion by treating this as a spherical cap, determining the cap angle, θ, from the fraction of the total circumference enclosed by the anterior segment, hence the cap radius, r, as equal to $R\sin\theta$ and the height of the cap, h, as equal to $R-\sqrt{(R^2-r^2)}$. The surface area of the cap is $= \pi(r^2+h^2)$ and hence the surface area of the retina is the total surface minus the cap area. Corneal thickness of control and mutant eyes was measured from H&E stained histological sections through the optic nerve using NDP.view2 software (Hamamatsu). Three measurements were made, in the centre of the cornea and at points equidistant between the centre and ciliary body. Cross sectional corneal images obtained by Anterior Segment OCT were used to

determine total central corneal thickness in scans which passed through the centre of the pupil. ImageJ software [59] was used to measure the linear distance between the anterior and posterior corneal surfaces in the central cornea. The measurements were obtained in pixels and the appropriate pixel to μm conversion factor was applied, relative to a 200 μm scale bar. One measurement was made for each scan of both eyes for each mouse. All measurement data is in S1 Dataset.

## RPE flat mounts

Immunohistochemistry on RPE flat mount was carried out as previously described [60]. Briefly, eyes were enucleated and the cornea, lens and retina removed. The RPE was dissected, radial incisions were made in the periphery to allow it to flatten, and it was fixed in cold (-20˚C) methanol for a minimum of 30 minutes. After rehydration in PBS RPE was blocked in whole mount buffer (PBS containing 3% Triton X-100, 0.5% Tween-20 and 1% bovine serum albumin (BSA)) which was also used for all washing and antibody incubation steps. After blocking RPE was incubated with primary antibodies overnight at 4˚C, washed and then incubated with secondary antibodies for two hours at room temperature and then washed, mounted in Prolong Gold (Thermo Fisher Scientific) and imaged as above.

## LacZ staining

For LacZ staining of cryosections slides were stained overnight in detergent buffer (0.1 M phosphate buffer pH7.3, 2 mM $MgCl_2$, 0.1% sodium deoxycholate and 0.02% NP-40 (IGEPAL CA-630)) containing 14.5 mM NaCl, 5 mM $K_3Fe(CN)_6$, 5 mM $K_4[Fe(CN)_6].3H_2O$ and 150 μg X-gal (5-bromo-4-chloro-3-indolyl-β-D-galactopyranoside) in a coplin jar at 37˚C protected from light. They were then washed twice in detergent buffer, post-fixed overnight in 4% paraformaldehyde (PFA) in PBS, counterstained with eosin and mounted using standard procedures. Brightfield images were acquired using a QImaging R6 colour CCD camera (QImaging, Surrey, BC, Canada) mounted on a Zeiss Axioplan II fluorescence microscope with Plan-neofluar or Plan Apochromat objectives (Carl Zeiss, Oberkocken, Germany). Image capture was performed using Micromanager (https://open-imaging.com/).

## Cell culture and transfection and stable cell line generation

NIH/3T3 and HEK293T cells were grown in DMEM (Thermo Fisher Scientific) supplemented with 10% fetal calf serum (FCS) and 1% penicillin/streptomycin. ARPE-19 cells were grown in DMEM:F12 (Thermo Fisher Scientific) supplemented with 10% FCS and 1% penicillin/streptomycin. All cells were grown in a humidified 5% $CO_2$ incubator at 37˚C. Cell transfection was carried out using a Neon Transfection System (Invitrogen) following the manufacturer's instructions except for the topology experiment described below. The optimal concentration of G418 antibiotic (Sigma-Aldrich, St. Louis, MO, USA) to kill ARPE-19 cells was determined by a kill curve using standard procedures (https://www.mirusbio.com/applications/stable-cell-line-generation/antibiotic-kill-curve) and G418 was used at a concentration of 0.8 mg/ml for generation and culture of stable cell lines. For generation of stable ARPE-19 cell lines, HEK293T cells were transiently transfected with pGAG/POL (gift from Tannishtha Reya (Addgene plasmid #14887) and pMD2.G (gift from Didier Trono (Addgene plasmid #12259) plus pQCXIN-BirA plasmids using Fugene HD (Promega, Madison, WI, USA). Virus-containing medium was collected, filtered through a 0.4 μm filter and mixed with an equal volume of fresh culture medium. Polybrene (Thermo Fisher Scientific) was added to 4 μg/ml and virus added to sub-confluent ARPE-19 cells. 12 hours later, medium was replaced. 48 hours after infection, stable expressing pools were selected using medium containing 0.8 mg/ml G418 (Thermo Fisher Scientific).

## Plasmids

To generate retroviral constructs for BioID experiments, the BirA-R118G open reading frame from the BirA(R118G)-HA destination vector (gift from Karl Kramer (plasmid # 53581, http://www.addgene.org) was subcloned into the retroviral vector pQCXIN (Clontech) to generate either N-terminal (pQCXIN-BirA-Myc-N) or C-terminal (pQCXIN-BirA-Myc-C) vectors. Full-length *Tmem98* without the stop codon was amplified by polymerase chain reaction (PCR) with the following primers: 5'- GGTTCCGTACGATGGAGACTGTGGTGATCGTC-3' and 5'- GGTTCGGATCCAATGGCCGACTGTTCCTGCAGG-3' which placed a BsiWI site at the 5' end and a BamHI site at the 3' end. This was cloned into the BsiWI and BamHI sites of the expression vector pQCXIN-BirA-Myc-C so that a BirA* domain was fused to the C-terminal of TMEM98. Other *Tmem98* expression constructs were made using the Gateway Cloning System (Thermo Fisher Scientific). The *Tmem98* open reading frame without the initiating ATG was amplified by PCR using primers 5'- CACCGAGACTGTGGTGATCGTCG-3' and 5'- AATGGCCGACTGTTCCTGCAG-3' and cloned into pENTR/D-TOPO (Thermo Fisher Scientific) and subsequently into pcDNA6.2/N-EmGFP-DEST (Thermo Fisher Scientific) to create GFP-Tmem98-V5, fusing a GFP tag at the N-terminal end of TMEM98 and a V5 tag at the C-terminal end. The *Tmem98* open reading frame with the initiating ATG was amplified by PCR using the primers 5'- CACCATGGAGACTGTGGTGATCGTCG-3' and 5'- AATGGCCGACTGTTCCTGCAG-3' and cloned into pENTR™/D-TOPO (Thermo Fisher Scientific) and subsequently into pcDNA-DEST40 (Thermo Fisher Scientific) to create TMEM98-V5, fusing a V5 tag at the C-terminal end of TMEM98. MYC-MYRF-FLAG and MYC-MYRFΔ19-FLAG have been described [39].

## Immunocytochemistry

Cells were grown on coverslips and fixed in 4% PFA/PBS for 10 minutes at room temperature, washed with TBS containing 0.1% Triton X-100 (TBSTx). Cells were blocked in 10% DS in TBSTx for one hour at room temperature. Primary antibodies diluted in TBSTx/1% DS were then added and incubated for one hour at room temperature or 4˚C overnight. Following washing with TBSTx all cells were incubated with Alexa Fluor secondary antibodies (Invitrogen) diluted 1:500 in TBSTx/1% DS at room temperature for one hour. Cells were then washed with TBSTx, incubated with 4′,6-diamidino-2-phenylindole (DAPI) at 2 μg/ml for five minutes, washed again with TBSTx and then mounted on slides in Prolong Gold (Thermo Fisher Scientific). Confocal images were acquired and processed as described above.

## Topology of TMEM98

GFP-Tmem98-V5 was transiently transfected into NIH/3T3 cells using Lipofectamine 2000 (Thermo Fisher Scientific) following the manufacturer's protocol and cells grown overnight in chamber slides. For immunofluorescence on permeabilised cells the cells were fixed in 4% PFA/PBS for 10 minutes, washed with TBSTx at room temperature (RT) followed by blocking in DS in TBSTx for 30 minutes and then incubated in primary antibodies diluted in TBSTx with 10% DS for one hour at 4˚C. For immunofluorescence on non-permeabilised cells the cells were washed with ice-cold TBS, blocked in TBS with 10% DS for 10 minutes followed by incubation with primary antibodies diluted in TBS with 10% DS for one hour, washed with TBS all at 4˚C and then fixed as described above. Subsequently, after washing with TBSTx, all cells were incubated with Alexa Fluor secondary antibodies (Invitrogen) diluted 1:1000 in TBSTx with 10% DS at room temperature for one hour. Following washing with TBSTx the chamber was removed and coverslips were mounted on slides in Prolong Gold (Thermo Fisher Scientific) and confocal images acquired and processed as described above.

## Fractionation of ARPE-19 cells and Western blotting

ARPE-19 cells were separated into cytoplasmic, membrane/organelle and nuclear/cytoskeletal fractions using the Cell Fractionation Kit (Cell Signaling Technology, Danvers, MA, USA) following the manufacturer's instructions. For Western blotting, fractions or equal amounts of protein lysates were separated on either 4–12% or 12% NuPAGE Bis-Tris gels (Thermo Fisher Scientific) depending on the size of the protein of interest and transferred to polyvinylidene difluoride (PVDF) or nitrocellulose membranes. Membranes were blocked for one hour at room temperature in SuperBlock T20 (TBS) Blocking Buffer (Thermo Fisher Scientific) and incubated with primary antibodies for one hour at room temperature or overnight at 4˚C in blocking buffer with shaking. Following washing with TBST membranes were incubated with ECL horse radish peroxidase (HRP)-conjugated secondary antibodies (GE Healthcare, Chicago, IL, USA) diluted 1:5000 in blocking buffer for one hour at room temperature, washed with TBST and developed using SuperSignal West Pico PLUS (Thermo Fisher Scientific).

## BioID experiment and mass spectrometry

ARPE-19 stable cell lines expressing TMEM98 tagged with BirA* fusion protein or BirA* alone were cultured in three 15 cm plates until 70–80% confluent. The culture medium was then replaced with medium containing 50 mM biotin and cultured for a further 24 hours when the cells were lysed using RIPA Buffer (Cell Signaling Technology) with 1 μM phenylmethylsulfonyl fluoride (PMSF) (Sigma-Aldrich) and Complete Protease Inhibitor Cocktail (Roche, Basel, Switzerland) following the manufacturer's instructions. Lysates were diluted in RIPA buffer with 1 μM PMSF and Complete Protease Inhibitor Cocktail (Roche) to a concentration of 1.68 mg/ml and 500 μl was used for capture of biotinylated proteins on streptavidin beads and subsequent mass spectrometry analysis. Streptavidin beads were diluted in IP buffer so that the equivalent of 5 beads were used per sample. Beads were transferred using a Thermo Kingfisher Duo into 500 μl lysate and incubated for one hour with mixing. All steps are at 5˚C unless otherwise stated. Beads were then transferred for two washes in IP buffer and three washes in TBS (300 μl each). After transfer into 100 μl 2M urea, 100 mM Tris, 1 mM DTT containing 0.3 μg trypsin per sample, beads were incubated at 27˚C for 30 minutes with mixing to achieve limited proteolysis. The beads were then removed and tryptic digest of the released peptides was allowed to continue for 9 hours at 37˚C. Following this, peptides were alkylated by adding iodoacetamide to 50 mM and incubating at room temperature for 30 minutes. Finally, peptides were acidified by addition of 8 μl 10% TFA. An estimated 10 μg of the resulting peptide solution was loaded onto an activated (20 μl methanol), equilibrated (50 μl 0.1% TFA) C18 StAGE tip, and washed with 50 μl 0.1% trifluoroacetic acid (TFA). The bound peptides were eluted into a 96-well plate (Axygen, Corning Inc., Corning, NY, USA) with 20 μl 80% acetonitrile (ACN), 0.1% TFA and concentrated to less than 4 μl in a vacuum concentrator. The final volume was adjusted to 15 μl with 0.1% TFA. Online LC was performed using a Dionex RSLC Nano (Thermo Fisher Scientific). Following the C18 clean-up, 5 μg peptides were injected onto a C18 packed emitter and eluted over a gradient of 2%-80% ACN in 48 minutes, with 0.5% acetic acid throughout. Eluting peptides were ionised at +2.2kV before data-dependent analysis on a Thermo Q-Exactive Plus. MS1 was acquired with mz range 300–1650 and resolution 70,000, and top 12 ions were selected for fragmentation with normalised collision energy of 26, and an exclusion window of 30 seconds. MS2 were collected with resolution 17,500. The AGC targets for MS1 and MS2 were 3e6 and 5e4 respectively, and all spectra were acquired with 1 microscan and without lockmass. Finally, the data were analysed using MaxQuant (v 1.5.6.5) (max planck institute of biochemistry) in conjunction with uniprot human reference proteome release 2016_11 (https://www.uniprot.com), with match between

runs (MS/MS not required), LFQ with 1 peptide required, and statistical analyses performed in R (RStudio 1.1.453 / R x64 3.4.4) (https://rstudio.com) using Wasim Aftab's LIMMA Pipeline Proteomics (https://github.com/wasimaftab/LIMMA-pipeline-proteomics) implementing a Bayes-moderated method [61]. The mass spectrometry proteomics data have been deposited to the ProteomeXchange Consortium (http://www.proteomexchange.org/) via the PRIDE [62] partner repository with the dataset identifier PXD017091.

## Co-immunoprecipitation

HEK293T cells cultured in 100 mm dishes were transfected with 5 μg of each expression plasmid as indicated. After 24 hours the cells were lysed using Cell Lysis Buffer (Cell Signaling Technology) with 1 μM PMSF and Complete Protease Inhibitor Cocktail (Roche) following the manufacturer's instructions. Lysates were incubated with anti-V5 agarose affinity gel (Sigma-Aldrich) at 4˚C overnight with agitation. The agarose affinity gel was then washed three times with Cell Lysis Buffer (Cell Signaling Technology) and bound proteins eluted with NuPAGE LDS Sample Buffer (Thermo Fisher Scientific) with NuPAGE Sample Reducing Agent (Thermo Fisher Scientific) by incubating at 70˚C for 10 minutes and separated on 4–12% NuPAGE Bis-Tris gels (Thermo Fisher Scientific) along with 5% of input samples and analysed by Western blotting as described above.

## Statistics

Statistical analysis was carried out using the program Graphpad Prism (Graphpad Software, San Diego, CA, USA). The statistical test used is indicated in the text. A value of $P<0.05$ was considered significant.

## Supporting information

**S1 Table. Genotyping primers.**
(DOCX)

**S1 Dataset. Eye measurements.**
(XLSX)

**S2 Dataset. ERG data.**
(XLSX)

**S1 Fig. *Tmem98* is expressed in the adult RPE, ciliary body and iris.** (A) LacZ staining of wild-type (male) and *Tmem98*$^{tm1b/+}$ (female) adult albino eyes collected at 5 weeks. The expression pattern of *Tmem98* is indicated by the blue staining for the reporter knockout allele *Tmem98*$^{tm1b}$. Eyes were enucleated and fixed in 4% PFA/PBS for an hour, rinsed in PBS and washed three times in detergent buffer (0.1 M phosphate buffer pH7.3, 2 mM $MgCl_2$, 0.1% sodium deoxycholate and 0.02% NP-40 (IGEPAL CA-630)). The eyes were then stained in detergent buffer containing 14.5 mM NaCl, 5 mM $K_3Fe(CN)_6$, 5 mM $K_4[Fe(CN)_6].3H_2O$ and 150 μg X-gal (5-bromo-4-chloro-3-indolyl-β-D-galactopyranoside) at 37˚C protected from light, washed twice in detergent buffer, post-fixed overnight in 4% PFA/PBS, rinsed in PBS and photographed. (B) Cryosections of LacZ-stained *Tmem98*$^{tm1b/+}$ (male) adult albino eye collected at 6 weeks show that *Tmem98* is strongly expressed in the RPE, ciliary body and iris. There is also some punctate staining in the lens and choroid. Cryosections were prepared as described in Materials and Methods and coverslips mounted in Vectashield (Vector Laboratories) prior to brightfield imaging. Scale bars represent 1 mm (A) and 100 μm (B).
(TIF)

**S2 Fig.** ***Tmem98*^*tm1c/tm1c*^ **mice have normal eyes.** Shown are slit-lamp pictures of eyes from two wild-type mice (top row), two *Tmem98*^*tm1c/tm1c*^ mice (middle row) and two *Tmem98*^*tm1c/tm1c*^; Tyr-Cre mice (bottom row). The mice in the top row are a female on the left and a male on the right at 11 weeks of age. The mice in the middle and bottom rows are female 9 week old litter-mates. Mice homozygous for the floxed conditional allele *Tmem98*^*tm1c*^ have eyes of normal size, whereas the eyes of the *Tmem98*^*tm1c/tm1c*^; Tyr-Cre mice are enlarged.
(TIF)

**S3 Fig. Tyr-Cre is expressed in the RPE, ciliary body and iris.** Cryosections of adult R26MTMG; Tyr-Cre eye showing that Cre is expressed in the RPE, ciliary body and iris and not in the neural retina (denoted by a white bar), lens or elsewhere in the eye. Tomato fluorescent protein (red) and green fluorescent protein (green). Scale bars represent 100 μm.
(TIF)

**S4 Fig. Ultrastructural analysis of the sclera.** Eyes from four month old male littermate mice were fixed overnight in 3% glutaraldehyde in cacodylate buffer at 4˚C then post-fixed in 1% osmium tetroxide for two hours at 4˚C. After dehydration through ascending grades of alcohol and propylene oxide they were impregnated with TAAB Embedding Resin (medium grade premix) and cured for 24 hours. Ultrathin sections were stained with uranyl acetate and lead citrate and viewed on a JEOL JEM 1200 EX2 transmission electron microscope fitted with an AMT Digital Camera using the AMTv600 image capture software. There does not appear to be any difference in the collagen bundle structure between the control *Tmem98*^*tm1c/1d*^ (left) and mutant *Tmem98*^*tm1c/tm1d*^; Tyr-Cre (centre and right). Scale bars represent 2 μm.
(TIF)

**S5 Fig. Eye shape is elongated when *Tmem98* is knocked-out in the eye by P6.** H&E stained head sections are shown. (A) Sections through the eye for one control *Tmem98*^*tm1c/tm1c*^ and three mutant *Tmem98*^*tm1c/tm1c*^; Tyr-Cre E17.5 embryos. The eye shape is noticeably elongated in one of the *Tmem98*^*tm1c/tm1c*^; Tyr-Cre embryos (top right) compared to the control (top left). (B) Sections through the eye for control *Tmem98*^*tm1c/+*^; Tyr-Cre (left) and mutant *Tmem98*^*tm1c/tm1c*^; Tyr-Cre P6 mice. The posterior segment of the mutant *Tmem98*^*tm1c/tm1c*^; Tyr-Cre eye is expanded and the retinal layers are thinner compared to the control. For P6 samples mice were culled and following decapitation and removal of the lower jaw heads were fixed in Davidson's fixative at 4˚C. Otherwise they were processed as described for embryos in Materials and Methods except that they were sectioned at 16 μm. Scale bars represent 250 μm (A) and 500 μm (B).
(TIF)

**S6 Fig. The predominant RPE *Myrf* transcript splice form lacks exon 19.** Shown is RT-PCR analysis of RPE collected from two C57BL/6 mice. The forward primer used was from exon 17 (5'-TAGCTCTGGTGGTGGTCATG-3') and the reverse primer spanned the exon 20/21 boundary (5'-GTAACCAGCAGCAAAGAGGG-3'). The predicted RT-PCR product sizes are 268 bp if exon 19 is included and 187 bp if exon 19 is excluded. The predominant splice form present in RPE is 187 bp (arrowed) which lacks exon 19. This was confirmed by sequencing. The sizes of the DNA fragments in the marker lane (M) are indicated to the left. The lane labelled with dash is a no template control. RNA was prepared using an RNeasy Plus Micro kit (Qiagen) following the manufacturer's instructions and first strand cDNA was prepared using a GoScript Reverse Transcription System (Promega).
(TIF)

**S7 Fig. Mutations H196P and I135T of TMEM98 do not affect its ability to inhibit MYRF self-cleavage.** (A) ARPE-19 cells were transiently transfected with TMEM98$^{H196P}$-V5 alone (top) or with MYC-MYRF-FLAG (bottom) and immunostained with anti-V5 (magenta), anti-MYC (Cell Signaling Technology, 2278) (red) and anti-FLAG (Biolegend, 637302) (green) antibodies as indicated. DAPI staining is in blue. When MYC-MYRF-FLAG is co-transfected with TMEM98$^{H196P}$-V5 it remains intact and colocalises with TMEM98-V5 in the membrane. The TMEM98$^{H196P}$-V5 construct was made in the same way as the TMEM98-V5 construct except that *Tmem98* open reading frame with the initiating ATG was amplified from cDNA isolated from *Tmem98$^{H196P/H196P}$* mice. (B) ARPE-19 cells were transiently transfected with TMEM98$^{I135T}$-GFP alone (top) or with MYC-MYRF-FLAG (bottom) and immunostained with anti-MYC (Cell Signaling Technology, 2276) (magenta) and anti-FLAG (Cell Signaling Technology, 2368) (red) antibodies as indicated. TMEM98$^{I135T}$-GFP is in green and DAPI staining is in blue. When MYC-MYRF-FLAG is co-transfected with TMEM98$^{I135T}$-GFP it remains intact and colocalises with TMEM98$^{I135T}$-GFP in the membrane. To make the TMEM98$^{I135T}$-GFP construct the *Tmem98* open reading frame with the I135T missense mutation was amplified by PCR using the primers 5'- GGGAGATCTCCCGGCATGCCCTGCTGCT GG-3' and 5'- CCCACCGGTATGGCCGACTGTTCCTGCAG -3' and cloned into the BglII and AgeI sites of pEGFP-N1 (BD Biosciences Clontech). Scale bars represent 20 μm. (TIF)

**S8 Fig. Western blot analysis of ARPE-19 subcellular fractions.** Uncropped images of the Western blots used to make Fig 6A. (TIF)

**S9 Fig. Western blot analysis of the co-immunoprecipitation experiment using tagged TMEM98 and MYRF constructs.** Uncropped images of the Western blots used to make Fig 7A. The antibodies used are indicated to the right of the images. (TIF)

## Acknowledgments

The authors thank IGMM scientific support services, Craig Nicol and Conor Warnock for help with photography, the IGMM Advanced Imaging Resource for help with imaging and Edinburgh University Bioresearch and Veterinary Services for animal husbandry. We thank Amy Findlay and Chloe Stanton for help with anterior segment OCT. The authors are very grateful to Michael Wegner for the gift of anti-MYRF (N-terminal) antibody, Ben Emery and Biliana Veleva-Rotse for the gifts of anti-MYRF (C-terminal) antibody and MYRF expression constructs and Ben Emery for helpful comments on the manuscript. Also Tannishtha Reya, Didier Trono and Karl Kramer for the gifts of plasmids.

## Author Contributions

**Conceptualization:** Sally H. Cross, Ian J. Jackson.

**Data curation:** Sally H. Cross, Lisa Mckie, Ian J. Jackson.

**Formal analysis:** Sally H. Cross, Jimi Wills, Ian J. Jackson.

**Funding acquisition:** Robert E. MacLaren, Ian J. Jackson.

**Investigation:** Sally H. Cross, Lisa Mckie, Toby W. Hurd, Sam Riley, Jimi Wills, Alun R. Barnard, Fiona Young.

**Methodology:** Sally H. Cross, Lisa Mckie, Ian J. Jackson.

**Project administration:** Sally H. Cross, Ian J. Jackson.

**Supervision:** Sally H. Cross, Ian J. Jackson.

**Validation:** Sally H. Cross, Ian J. Jackson.

**Visualization:** Sally H. Cross, Ian J. Jackson.

**Writing – original draft:** Sally H. Cross, Ian J. Jackson.

**Writing – review & editing:** Sally H. Cross, Toby W. Hurd, Jimi Wills, Fiona Young, Robert E. MacLaren, Ian J. Jackson.

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
