## [Decision Letter · Decision Letter 0]

4 Feb 2020

Dear Dr Cross,

Thank you very much for submitting your Research Article entitled 'The nanophthalmos protein TMEM98 inhibits MYRF self-cleavage and is required for eye size specification' to PLOS Genetics.

The manuscript was fully evaluated at the editorial level and by two independent peer reviewers. As you will see, both reviewers are positive about the quality, rigor, and extent of the work; both reviewers also note that the potential impact of the work is primarily confirmatory.

The manuscript and the reviews have now been considered by members of the editorial board. Overall, we agree with the reviewers' comments, and ask that you carry out minor revisions to the presentation that address the concerns noted below.

We therefore ask you to modify the manuscript according to the review recommendations before we can consider your manuscript for acceptance. Your revisions should address the specific points made by each reviewer.

[LINK]

Yours sincerely,

Gregory S. Barsh

Editor-in-Chief

PLOS Genetics

Gregory Copenhaver

Editor-in-Chief

PLOS Genetics

Reviewer's Responses to Questions

**Comments to the Authors:**

Reviewer #1: Cross et al. presented the ocular phenotypes of a mouse when Tmem98 was knocked out in RPE and provided further evidence of the interaction of TMEM98 and MYRF in this well written manuscript. The authors presented strong rationale to investigate the function of TMEM98 given its association with eye size in human patients. On the first glance, it is confusing as the mutations in TMEM98 cause nanophthalmos is human patients, the overwhelming phenotype in mouse with RPE specific Tmem98 KO is axial enlargement of the eye. However, it is not uncommon at all, even just in human patients. Some mutations in the same gene can cause opposite phenotypes. This perhaps should be added to the discussion. In addition, mouse models often fail to recapitulate human diseases. The evidence of interaction between TMEM98 and MYRF is convincing, yet not terribly exciting given the recent publication in this specific journal by Garnai et al. (PLoS Genet. 2019 May; 15(5): e1008130). The idea of interaction of RPE with sclera in the setting of eye development and size is very intriguing and brought out and studied by the authors, but it falls short (see comments below), which is somewhat disappointing.

Line 100-102: "thus identifying a pathway..." seams a bit strong as this has been previously shown by Garnai et al. Adding further evidence should be considered.

Line 114: "the eyes of majority (5/6) KO mice" is not clear. When the mouse had elongated eye, was it for both eyes? 5/6 refers to 5 out of 6 eyes or mice? I suppose it means 5/6 mice. If so, was the elongation similar in both eyes from the same mouse?

line 149-153: given the lack of quantification and relatively small number of animals, it is hard to be totally convinced on the reduction of autofluorescence. Similar to the comments above, were two eyes from the same mouse similar?

Line 173-175: the sclera appears very thin in mutant mice. Although the authors stated that they did not observe any obvious abnormalities by TEM, was there some subtle disorganized structure as shown in S4? Was TEM looking at collegian fibrils good enough or other studies should be conducted? It is a missed opportunity to not study sclera in details. Rather, the authors went on to show ERG changes. Such change should not be a surprise. This does not add much significance of this study, hence it should be moved to supplemental materials.

Line 336-338: "numerous studies", references should be given. This is very intriguing and should be further studied.

Lone 384-386: It indeed would provide ultimate evidence if Myrf KO could rescue the enlarged eye phenotype of Tmem98 KO. Do the authors have the plan to make the double KO mouse? Any data? As the phenotype can be characterized in embryos, it may not be a very difficult and time consuming experiment.

Line 411: R26MTMG mice: delete mice.

Figure 2: Did authors measure white to white distance? What about corneal thickness by OCT? It would add more evidence that anterior segments are normal.

Fig 3. The emphasis is on the retina thinning. However, the other structures, such as choroid and sclera are vastly abnormal too, which could play a important role in globe elongation.

Fig 4. should be moved to supplemental materials.

Fig. 6B. Information appears scant. N of ? Difference between top and bottom panels?

S1 Fig. it appears Tmem98 is also expressed in lens and choroid???

S3 Fig. legend: RPE should be changed to iris.

S5 Fig. Given the fact the contact lenses have to be changed because of elongation of the eye in mutant mice, this reviewer questions the reliability of comparison between control and mutant. Why there is such large variation in temporal retina in both control and mutant mice? One male and two females at age between 10-14 weeks were used. Sex and age (4 weeks difference) could play a role??? Consider remove this figure as it does not add much more information.

S6 Fig: lens in the mutation in middle and bottom panel is larger than corresponding control in this figure, which may indicate the pariticular section shown here is more central, wihle the control is not. This could result in larger appearance of the eye, as often seen in histology.

Reviewer #2: This is a well written study of a new animal model, a conditional knockout of TMEM98 in the retinal pigment epithelium, which gives insight into one possible contributory factor myopia. That is important because myopia is a major clinical burden and is increasing in the human population for reasons only partially understood; and because this finding gives rare functional insight into the mechanism by which SNPs detected in a GWAS contribute to a human phenotype. There were a few sentences I could improve for readability - the authors don't seem to like using commas much - but I cannot suggest improvements that are beyond the brief of a copy editor.

In an ideal world it would have been nice to explore why these mutations cause different phenotypes in mice and humans, and why some TMEM98 variants increase eye size and others reduce it. Their use of the word "unexpectedly" in describing what they found serves to underline this complex question. For the latter point they hypothesize in discussion that the human nanophthalmos-causing mutations are gain of function, and say as evidence that a compound het mouse with complete knockout and missense had no phenotype, so each compensates for the other. There is no reference for this missing bit of data - was it in the previous paper? Anyway that does not explain why key missesne variants or deletion of the C terminus cause nanophthalmos in humans, while complete knock-out increases the posterior eye size in mice.

But I assume these puzzles will be the target of their future work. As a self-contained project the analysis of this mouse conditional knockout is beautifully documented, thoroughly described, and does contain original and important new findings. The abstract is slightly confusing - if I read it correctly the novelty is not in finding that TMEM98 inhibits the self-cleavage of MYRF, documented in ref 49, but that this interaction is so crucial in determining eye size.

**Have all data underlying the figures and results presented in the manuscript been provided?**

Reviewer #1: Yes

Reviewer #2: Yes

PLOS authors have the option to publish the peer review history of their article (what does this mean?). If published, this will include your full peer review and any attached files.

Reviewer #1: No

Reviewer #2: Yes: Chris Inglehearn

---

## [Editor Report · Decision Letter 1]

6 Mar 2020

Dear Dr Cross,

We are pleased to inform you that your manuscript entitled "The nanophthalmos protein TMEM98 inhibits MYRF self-cleavage and is required for eye size specification" has been editorially accepted for publication in PLOS Genetics. Congratulations!

Yours sincerely,

Gregory S. Barsh

Editor-in-Chief

PLOS Genetics

Gregory Copenhaver

Editor-in-Chief

PLOS Genetics

Comments from the reviewers (if applicable):

**Data Deposition**

http://datadryad.org/submit?journalID=pgenetics&manu=PGENETICS-D-19-02092R1

**Press Queries**

---

## [Editor Report · Acceptance letter]

20 Mar 2020

PGENETICS-D-19-02092R1 

The nanophthalmos protein TMEM98 inhibits MYRF self-cleavage and is required for eye size specification 

Dear Dr Cross, 

We are pleased to inform you that your manuscript entitled "The nanophthalmos protein TMEM98 inhibits MYRF self-cleavage and is required for eye size specification" has been formally accepted for publication in PLOS Genetics! Your manuscript is now with our production department and you will be notified of the publication date in due course.

With kind regards,

Matt Lyles

PLOS Genetics

On behalf of:
